# Bionic artificial skin with a fully implantable wireless tactile sensory system for wound healing and restoring skin tactile function

Kyowon Kang[1,18], Seongryeol Ye[2,3,18], Chanho Jeong[4,18], Jinmo Jeong [5,6,18], Yeong-sinn Ye[4], Jin-Young Jeong[5], Yu-Jin Kim[2], Selin Lim[1,7], Tae Hee Kim[2,8], Kyung Yeun Kim[9,10], Jong Uk Kim[11], Gwan In Kim[1,12], Do Hoon Chun[1], Kiho Kim[1], Jaejin Park[1], Jung-Hoon Hong[1], Byeonghak Park[13], Kyubeen Kim[1], Sujin Jung[1], Kyeongrim Baek[1], Dongjun Cho[1], Jin Yoo[2], Kangwon Lee[3,14], Huanyu Cheng [15], Byung-Wook Min [1], Hyun Jae Kim [1], Hojeong Jeon[9,16], Hyunjung Yi [5,17] ✉, Tae-il Kim [4] ✉, Ki Jun Yu [1,7] ✉ & Youngmee Jung [2,7] ✉

Tactile function is essential for human life as it enables us to recognize texture and respond to external stimuli, including potential threats with sharp objects that may result in punctures or lacerations. Severe skin damage caused by severe burns, skin cancer, chemical accidents, and industrial accidents damage the structure of the skin tissue as well as the nerve system, resulting in permanent tactile sensory dysfunction, which significantly impacts an individual's daily life. Here, we introduce a fully-implantable wireless powered tactile sensory system embedded artificial skin (WTSA), with stable operation, to restore permanently damaged tactile function and promote wound healing for regenerating severely damaged skin. The fabricated WTSA facilitates (i) replacement of severely damaged tactile sensory with broad biocompatibility, (ii) promoting of skin wound healing and regeneration through collagen and fibrin-based artificial skin (CFAS), and (iii) minimization of foreign body reaction via hydrogel coating on neural interface electrodes. Furthermore, the WTSA shows a stable operation as a sensory system as evidenced by the quantitative analysis of leg movement angle and electromyogram (EMG) signals in response to varying intensities of applied pressures.

Tactile sense, one of the five fundamental senses (touch, sight, smell, hearing, and taste), is vital for organisms to interact with their surroundings and respond rapidly and effectively to various external stimuli. Mechanoreceptors and tactile neurons, located in the dermis layer of the skin, change their potential upon receiving external pressure and modulate the firing behavior of tactile neurons, converting the external pressure into electrical pulse signals with different frequencies, depending upon the intensity of the pressure. However, in cases of severe skin damage due to second or third-degree burns, chemical accidents, or industrial accidents, the consequences extend beyond structural impairment of the skin tissue, and tactile sensory dysfunction due to permanent damaged mechanoreceptors and associated nerves. In the United States alone, over 400,000 patients are suffering from burn-related injuries annually and more patients are considered to suffer from severe skin damage, highlighting the widespread nature of this issue and its impact on tactile loss globally[1]. Surgical interventions[2–4] (e.g., skin graft) or pharmacological treatments[5,6] are often employed to restore the skin or promote skin regeneration, whereas natural healing of severe scarring requires time and can be fatal because of infections in the worst case. Severely

damaged skin can be treated using hydrogel-based artificial skins, which enable layer-by-layer regeneration of the severely damaged skin and minimize scar formation by providing a microenvironment similar to the extracellular matrix (ECM) of the native skin on the wound site. Despite the advances in artificial skins based only on biomaterials, the restoration of tactile function remains a challenge because damaged mechanoreceptors and tactile neurons are unable to regenerate, even when the epidermis and dermis are successfully regenerated. As a result, patients with severe skin damage may experience permanent tactile sensory impairment, which significantly affects their daily lives.

Owing to the significant impact of impaired tactile sensation on the quality of daily life, various studies have been conducted to address issues, such as nerve stimulation, using an artificial neuromorphic tactile sensory system[7–11], which demonstrated the feasibility for replacement of tactile sensory function by enabling nerve stimulation through externally applied strain or pressure. Although previous researches have introduced devices capable of replacing tactile function[12], no single substrate integrated system that can accelerate regeneration of a severely damaged skin and substitute tactile function with minimized foreign body reactions has yet been reported[11,13–17]. Furthermore, the integration of each functional component of the device using external wires results in a bulky device and potential inclusion of noise. Successfully implementing an artificial tactile sensory system to replace damaged skin, with permanently impaired tactile function, requires not only stable operability as a device but also the ability to accelerate skin regeneration and ensure biocompatibility with minimal foreign body reactions for effective nerve stimulation. In addition, at the interface between the surrounding living skin tissue and the device, ECM-based natural hydrogels, such as collagen and fibrin, not only enhance the biocompatibility of the sensor but also serve as a buffer to help regenerate the complete skin tissue. Introducing foreign substances into the body inevitably triggers a foreign body reaction, which is a complex protective mechanism involving acute immune responses and fibrosis to neutralize external threats. After implanting a device into the body, an initial inflammatory response degrades the inserted device, and then a foreign substance covers the fibrotic tissues, thereby interrupting biosignal recording or electrical stimulation. Therefore, for the electrodes of the implantable device, minimizing fibrosis and ensuring noncytotoxicity are essential for realizing a stable and long-term operation of the implanted devices.

In this article, we introduce a bionic artificial skin with a fully implantable wireless tactile sensory system for accelerating regeneration of severely damaged skin and restoring its tactile function with minimized foreign body reaction. The artificial skin facilitates an accelerated layer-by-layer skin regeneration, and the hydrogel-coated nerve interface minimizes foreign body reactions to realize an effective long-term nerve stimulation, while replacing the permanently damaged mechanoreceptors with a tactile sensor. The proposed tactile sensory skin system consists of an ECM-based artificial skin, a crack-based pressure sensor, a wireless-powered pressure-frequency modulation (WPPFM) circuit, neural interface electrodes, and multilayer encapsulation. The complete system, which integrates all these components on a single substrate, facilitates biomimetic tactile sensory recognition by transmitting the frequency-based action potential depending on the intensity of the applied pressure. The tactile sensor embedded in the artificial skin detects the intensity of the externally applied pressure via changes in the resistance[18–23]. The artificial skin surrounding the tactile sensor, composed of the main ECM components of the native skin, exhibits mechanical properties that are similar to those of the skin tissue and thus effectively accelerates skin regeneration and wound healing. Then, the WPPFM circuit converts the tactile signal (resistance) into a frequency-based sawtooth pulse signal (Δfreqeuncy) through a ring oscillator. This converted signal is transferred to the neural interface electrodes to stimulate the sciatic nerve, which correlates with leg movements[7,8,24]. The hydrogel coating

on the neural interface electrodes, whose mechanical properties are similar to those of the sciatic nerve, minimizes nerve damage due to friction occurring during movement and suppresses fibrosis, thereby effectively stimulating the nerve after long-term implantation. The entire device is multilayer-encapsulated to protect the device from biofluids for long-term operation as well as to protect the nearby tissues from the leakage current. The wireless-powered tactile sensory system embedded artificial skin (WTSA) is implanted to a rat model with severe skin damage, and the correlation between the pressure-induced pulse signal and leg movements is precisely analyzed by monitoring the leg movement angle and the corresponding electromyogram (EMG) signals.

## Results
### Overview of the WTSA
Figure 1a and Supplementary Fig. 1 show the structure and fabrication process of the integrated WTSA with five main parts; (i) artificial skin, (ii) crack-based tactile sensor, (iii) WPPFM circuit, (iv) hydrogel-coated neural interface electrodes, and (v) multilayered encapsulation for long-term implantation. To replace the permanently damaged mechanoreceptors, a crack-based tactile sensor, embedded in the artificial skin, is used to sense the externally applied pressure[19,25] (Supplementary Fig. 2). The cuboid structure at the front of the tactile sensor improves its accuracy and responsiveness by concentrating the applied force on the structure. When an external pressure is applied, structural deformation widens gap between the cracks on the Pt metal layer, thereby impeding the current flow and resulting in a drastic increase in resistance[18]. The WPPFM which operates by wireless power transfer, modulates the tactile signal in the form of resistance to sawtooth pulse signal with different frequencies. The interconnection between WPPFM and neural interface electrodes is placed on the leg joint area where continuous stretching occurs, by applying serpentine structured interconnection[26,27], the durability of interconnection is achieved, as shown in Supplementary Fig. 3. Then, the sciatic nerve is stimulated with the hydrogel-coated neural interface electrodes to minimize fibrosis after long-term implantation.

The proposed WTSA not only replaces the permanently damaged tactile senses but also accelerates full-thickness skin regeneration. Figure 1b shows the overall accelerated skin regeneration process of severely damaged skin using the artificial skin system. Artificial skin based on hydrogel mixture with collagen and fibrin promotes wound healing and full-thickness skin regeneration by providing a microenvironment similar to that of the ECM of the native skin to the severely damaged skin area. In the case of neural interfaces for electrical stimulation, fibrotic tissues generated between the electrode and nerve by the foreign body reaction after device insertion partially block the electrical stimulation path. Figure 1c presents the effectiveness of minimizing foreign body reactions on the neural interface electrodes by coating the hydrogel. Unlike the neutral interface electrodes without the hydrogel coating, the ones coated with the hydrogel enabled an effective nerve stimulation after long-term implantation by providing a hydrated environment between the nerve and electrodes, thereby minimizing fibrotic tissue formation.

Figure 1d and Supplementary Fig. 4 show the tactile signal conversion and electrical stimulation process of the implanted WTSA, which mimics the human tactile sensory system. When pressure is applied to severely damage skin area, where mechanoreceptors and tactile neurons are permanently damaged to lose tactile function, the resistance-based tactile sensor detects the applied pressure by deformation of microcracks on tactile sensor. Then WPPFM converts the tactile signal in the form of resistance to voltage difference via a voltage divider, and ring oscillator modulate the voltage difference into sawtooth pulse signal with different frequencies followed by sciatic nerve stimulation through hydrogel-coated neural interface electrodes[7,9,10]. The electrical stimulation of sciatic nerve triggers the

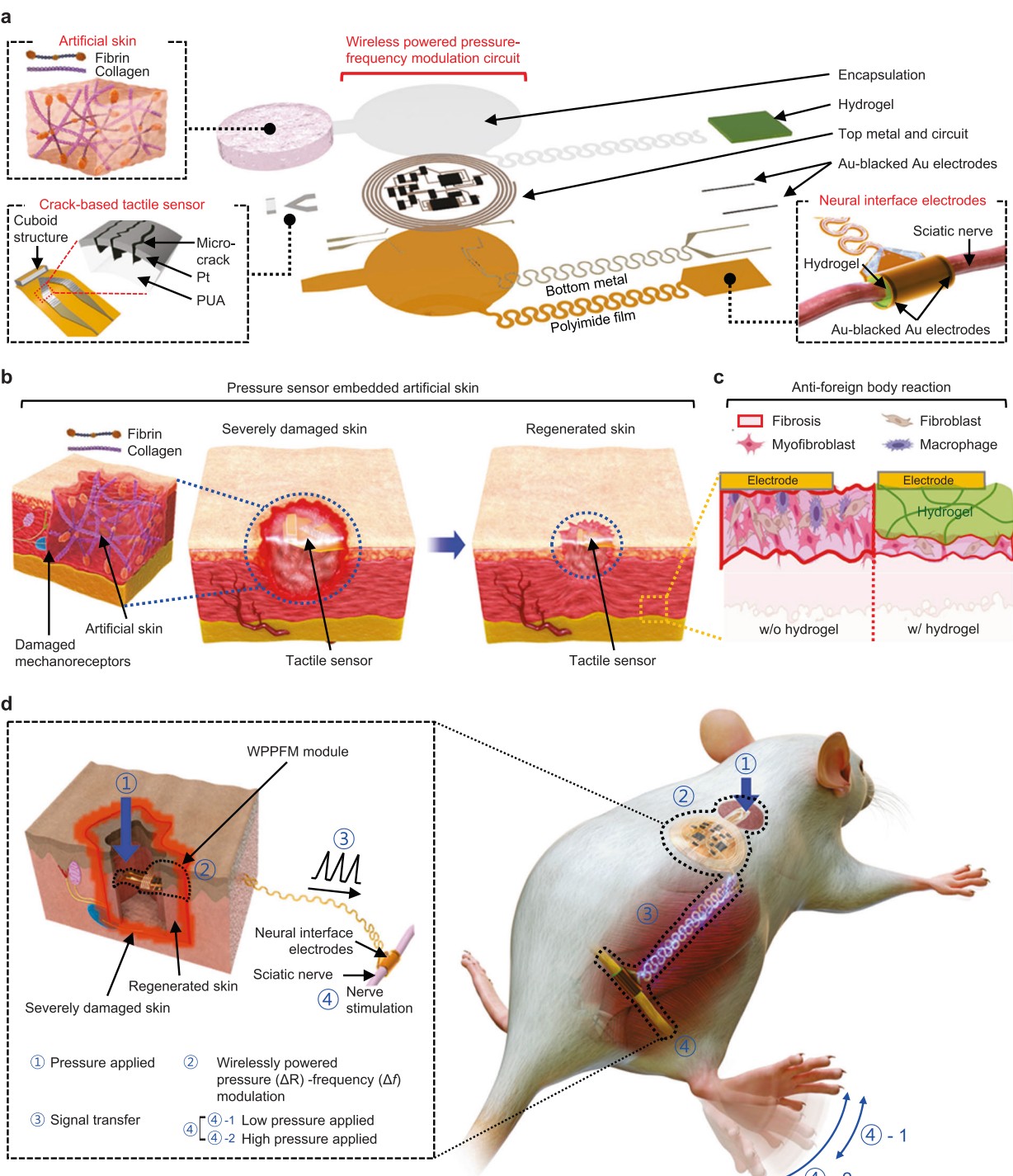

**Fig. 1 | Overall schematic of the WTSA and converted tactile signal transfer process to stimulate the sciatic nerve. a** An exploded view schematic of WTSA composed of artificial skin, crack-based tactile sensor, WPPFM circuit, neural interface electrodes and fibrin coating to minimize foreign body reaction. **b** Accelerated skin regeneration process through an ECM-based artificial skin composed of collagen and fibrin. **c** Suppression of the foreign body reactions by the hydrogel layer for an effective neural stimulation. Created with BioRender.com. **d** Schematic image of tactile signal modulation and signal transfer process for stimulating the sciatic nerve with respect to the different intensity of applied pressure. Left inset image with dashed box shows the magnified image of the implanted WTSA.

contraction of the associated muscles, resulting in leg movement with different angles depending on the intensity of applied pressure. When high pressure is applied, the resistance of the tactile sensor increases, leading to an increase in the frequency of the sawtooth pulse signal. This, in turn, induces stronger muscle contraction and greater leg movement.

**Tactile signal modulation and nerve stimulation with WPPFM**

Implantable devices require a wireless power transfer ability to control the devices and deliver power from the outside of the body. Thus, the implantation of additional bulky batteries is not required. Figure 2a shows the image of WTSA designed to restore the permanently dysfunctional tactile sensory, varying in response to the strength of the

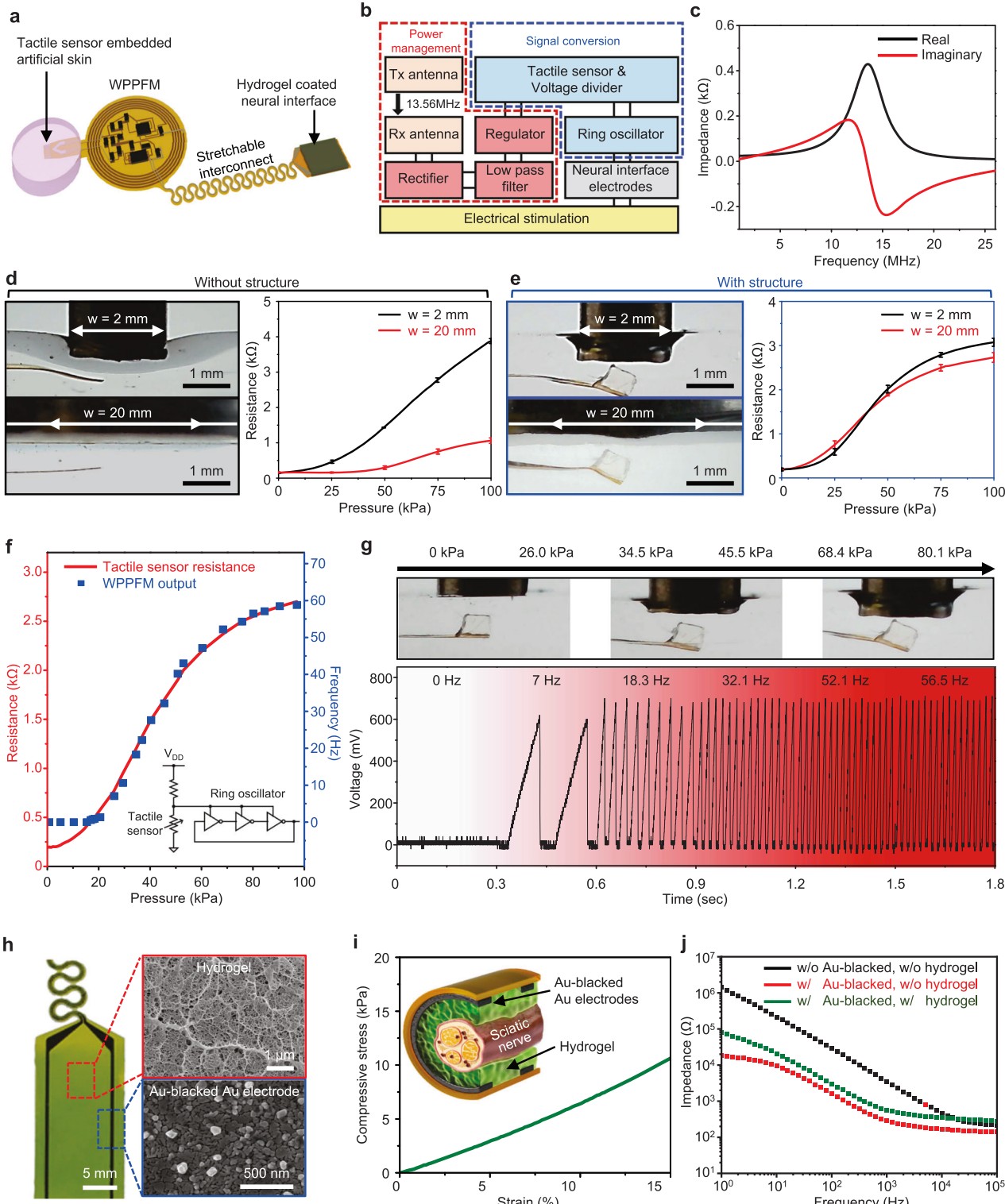

**Fig. 2 | Characteristics of the WPPFM and neural interface electrodes.**
**a** Schematic of WTSA. **b**, Block diagram of WPPFM and neural interface electrodes.
**c**, Wireless power transfer characteristic by matching the resonant frequency at
13.56 MHz. **d**, **e** The photographs (left) and the resistance change (right) while
pressing the tactile sensor with rod (diameter of 2 mm or 20 mm) according to the
existence of a cuboid structure (**d**; without cuboid structure, **e**; with cuboid
structure). Representative data points are presented whereas standard deviation
(SD) are shown as error bars (*n* = 3). **f**, **g** The resistance of tactile sensor (red) and the

frequency output of WPPFM (blue) according to the applied pressure (**f**). The tactile
modulated sawtooth pulse signal output from WPPFM according to the pressure
applied (**g**). **h** Photograph of hydrogel-coated neural interface electrodes. Inset
scanning electron micrographs images show the 3D scaffold structure of the
hydrogel (red) and the nanostructured surface of the Au-blacked Au electrodes
(blue). **i** Compressive stress–strain curve of the hydrogel. **j** Electrochemical impe-
dance spectra graph of bare Au electrode, Au-blacked Au electrode, Au-blacked Au
electrode with hydrogel.

applied pressure. Figure 2b and Supplementary Fig. 5 present the block diagram of WPPFM circuit and nerve stimulation which modulate the externally applied pressure to sawtooth pulse signal with different frequencies for electrical stimulation of the sciatic nerve. The receiver coil (Rx) is designed to resonate at 13.56 MHz, a commonly used frequency for general electronic devices[28–30] (Fig. 2c). Supplementary Fig. 6 shows 20 dB return loss(matching) at 13.56 MHz when the Rx is placed on the transmit coil (Tx), indicating that an efficient wireless power transfer is achieved at this frequency. The wirelessly transferred AC signal is rectified into a DC signal with ripples through full-bridge rectifier which is composed of four Schottky diodes and the ripples are moved by low-pass filter which is designed to have cutoff frequency at 338.6 Hz. To regulate the DC value constant while the transferred AC signal fluctuates due to the location difference between the transmitter and receiver coils, a linear DC regulator with a regulation voltage of 1.2 V is used, which successfully regulates the DC voltage to 1.2 V (Supplementary Fig. 7). The regulated voltage supplies lowered constant voltage (1.2 V) for device operation by voltage divider circuit which is composed of fixed resistor (R, 150 Ω) and resistance-based tactile sensor ($\Delta R_{pressure}$) resulting the resistance change of tactile sensor can be converted into voltage change ($1.2 \times \frac{\Delta R_{pressure}}{R + \Delta R_{pressure}}$). The output frequency of the ring oscillator varies depending upon the input voltage differences. Since, commercial ring oscillator offer a high output frequency, by connecting the three additional capacitors at the end of each three inverter connected in a loop, the time consumed to charge and discharge the three additional capacitor intentionally provide the time delay to each inverter of the ring oscillator, resulting in the low-frequency (<100 Hz) output which is appropriate to stimulate the nerve[7,31,32]. The sawtooth output signal with different frequencies with respect to the intensity of the applied pressure stimulates the sciatic nerve via hydrogel-coated neural interface electrodes.

The presence of the cuboid structured polyurethane acrylate (PUA) block located in front of the tactile sensor assists in concentrating the applied pressure force on the crack-based tactile sensor (Supplementary Fig. 8). Supplementary Fig. 9 shows a comparative finite element method (FEM) simulation analysis of the presence of the cuboid structured PUA block (500 μm height) when pressure is applied through narrow width (2 mm) and broad width (20 mm) rods. The deformed cross-sectional tactile sensor with the stress distribution illustrates the effect of the cuboid structure's existence. Without the cuboid structure, broad-area pressure generates uniform pressure over the sensor, resulting in shifting of the entire sensor while remaining flat. However, with the cuboid structure, even when broad pressure is applied, the pressure concentrates to the structure resulting in the bending of the sensor. Therefore, the tactile sensor with the cuboid structure (Supplementary Fig. 9; Blue box) effectively bends and shows almost identical stress distribution under localized and broadly applied pressure while tactile sensor without the cuboid structure (Supplementary Fig. 9; Black box) only bends when localized pressure is applied and merely bends under broad-area pressure. Figure 2d, e presents the tactile sensor characteristics, embedded in a skin-like soft PDMS, depending on the presence of the cuboid structure under a localized and broadly applied pressure up to 100 kPa, which corresponds well to the FEM simulation results. When the cuboid structure is absent, the tactile sensor shows a significant resistance change under a localized pressure, whereas a small resistance change is observed under a broadly applied pressure, indicating that the tactile sensor fails to effectively detect the broadly applied pressure. On the other hand, however, the tactile sensor with the cuboid structure, which concentrates the applied pressure on the tactile sensor, can detect both localized and broadly applied pressures, exhibiting nearly identical pressure-resistance curves. These results indicate that the presence of the cuboid structure at the front of the tactile sensor significantly improves the pressure-sensing property

under various pressure conditions, including localized and broad pressures. Supplementary Fig. 10 and Supplementary Note 1 theoretically demonstrate this effect. As shown, in the presence of the cuboid structure, there is more compression along with stress concentration in the surrounding areas, with the height of the cuboid structure being a key factor. Figure 2f, g shows the frequency of the signal output from WPPFM corresponds to the change in the resistance of the tactile sensor with respect to the applied pressure resulting the increment of the frequency from WPPFM output when the intensity of applied pressure increases. To accommodate the curved surface characteristics of the body, fine-tuning of the fixed resistance value of voltage divider (150 Ω) allows the avoidance of frequency output when the tactile sensor is attached on curved surfaces and slightly bent without external pressure. Tactile sensor and WPPFM circuit are multilayer-encapsulated with a Polydimethylsiloxane (PDMS)/Parylene C/$Al_2O_3$ (100 μm/2 μm/50 nm) hybrid layer for long-term operation after the implant[33]. The phosphate-buffered saline (PBS) acceleration test shows that Mg degrades after 248 h, 92 h, and 29.8 h under 70 °C, 80 °C, and 92 °C PBS solution, respectively, due to the failure of the encapsulation, indicating that the proposed multilayer encapsulation can survive for 429 days at a physiological temperature (37 °C), which can be calculated by the Arrhenius equation; $Lifetime = 1.533 \cdot 10^{-13} \cdot e^{-\frac{99909.629}{8.314 \cdot T}}$, where 99,909.629 is activation energy, 8.314 is the gas constant, and T is temperature[33–35] (Supplementary Fig. 11 and Supplementary Note 2).

To achieve a clinically long-term sustainable neural interface, mechanically and electrically stable neural electrodes with anti-foreign body reactions need to be developed[36]. Figure 2h presents neural interface electrodes based on Au-blacked Au electrode and a hydrogel layer. A genipin-cross-linked gelatin hydrogel with a suitable 3D scaffold was utilized as an intermediate layer between Au-blacked Au electrodes and the sciatic nerve. In our previous study, we optimized the genipin-crosslinked gelatin hydrogel with highly appropriate composition ratio for neural interface use in terms of mechanical and electrical properties[37]. Figure 2i shows the mechanical properties (-70.8 kPa) of 3D scaffold-based hydrogel, which closely match those of the sciatic nerve[38]. Moreover, the hydrogel has ion transport properties due to its high water absorbance, facilitating stable electrical signal transmission without inflammatory responses[39]. The measured conductivity of the hydrogel was 0.010188 ± 0.000212 S/cm, still showing 55% of the conductivity level in PBS (0.018478 ± 0.00019 S/cm). In terms of the impedance, the impedance of PBS and hydrogel at 1 kHz were 19.238 ± 2.787 Ω and 24.354 ± 0.193 Ω, respectively, which showed rather similar values than those in the high-frequency range (Supplementary Fig. 12). An effective electrical performance of the electrodes was achieved through the utilization of the Au-blacked electrodes. The electrodeposition of the Au-black onto the Au electrode formed an Au nanogranular surface, leading to an increase in the electrochemical surface area (ESA) of the electrode[40] (Fig. 2h). The increased ESA can reduce the impedance and increase the charge storage capacity (CSC)[41]. The CSC of the Au-blacked Au electrode was -0.900 mA/cm$^2$ at 0.15 V/s which was five times higher than that of the bare Au electrode (-0.176 mA/cm$^2$ at 0.15 V/s), showing a sufficient value for neural stimulation[40] (Supplementary Figs. 13 and 14). The neural interface electrodes were fabricated with Au-blacked Au electrodes and a hydrogel layer to minimize foreign body reactions and to enable an effective neural stimulation. Figure 2j shows the impedance plots of the bare electrode, Au-blacked Au electrodes, and hydrogel-coated Au-blacked Au electrodes. The impedance of the Au-blacked Au electrodes was 284.2 Ω at 1 kHz, -12 times lower than that of the bare electrodes (3481.0 Ω at 1 kHz). Furthermore, the impedance magnitude of the neural interface electrodes with hydrogel was 570.0 Ω at 1 kHz, slightly higher than that of the Au-blacked Au electrodes, but still six times lower than that of bare Au electrodes.

## Characterization and wound-healing efficacy of artificial skin

The artificial skin, in which collagen and fibrin were mixed, not only promoted the wound-healing process by providing microenvironments mimicking the ECM of the native skin but also exhibited mechanical properties similar to that of the native skin[42,43]. The artificial skin resisted physical deformations due to mechanical stress, such as tensile or compressive stresses, while temporarily replacing the severely damaged skin[44]. In the wound-healing processes, collagen helps fibroblasts to form the dermis and substitute the dead cells for new cells while providing the microstructure to support the elasticity of human skin[45,46]. Fibrin, the most abundant ECM protein during the initial hemostasis phase of wound healing, prevents excessive blood loss and provides a matrix to promote migration of the tissue repair cells[47]. Therefore, CFAS of the WTSA is expected to accelerate wound healing and skin regeneration by supporting differentiation and maintenance of migrated surrounding cells and newly formed tissues. Figure 3a shows the scanning electron microscope (SEM) image and surface morphology of CFAS with the fiber-based microporous structure to embed helping the interactions between cells involved in wound regeneration by providing microporous spaces where host cells can be effectively placed[48]. Figure 3b, c and Supplementary Fig. 15 present the viscoelastic behavior of the CFAS by measuring the mechanical properties. Rheology results over frequencies of about 100 Pa of storage modulus and 50 Pa of loss modulus indicating that CFAS has a suitable stiffness to keep the shape in the form of hydrogel maintaining the wet condition to accelerate skin regeneration[49]. In addition, the maximum resistance stress under the tensile stretching test was 25 kPa, demonstrating that the CFAS has enough viscoelasticity to be used as the temporal skin replacement[50] (Supplementary Fig. 16).

To investigate the wound-healing effect of CFAS, wound defects were locally treated with CFAS, tactile sensor, or CFAS with tactile sensor (CFAS@Tactile sensor), and untreated wounds (Defect) were used as the control group (Fig. 3d–h). The effect of CFAS to promote wound healing was significantly different on day 3 in CFAS and CFAS@Tactile sensor groups compared to the tactile sensor group. In addition, as shown in CFAS and CFAS@Tactile sensor groups of day 7, remaining CFAS promotes the formation of new tissues by providing sufficient moisture condition to the wound site. The CFAS showed a significant effect on the wound-healing state by providing the microenvironments to introduce surrounding cells and structural scaffold to the severely damaged skin compared to the tactile sensor group[51,52]. As a result of treating with CFAS, at day 21, the CFAS-treated groups (CFAS and CFAS@Tactile sensor) showed complete wound closing, while the tactile sensor group's scars slightly remained (Fig. 3d, e). These results indicate that the contraction of the wound would be inhibited by the structural obstruction of the tactile sensor resulting in deficient regeneration of the epidermal layer. In contrast, in the CFAS@Tactile sensor group, the presence of collagen and fibrin in the CFAS encourage the activity of myofibroblasts which establish a grip on the wound edges contracting themselves, leading to complete wound regeneration, regardless of the structural obstruction of tactile sensor[53]. In addition, the pressure-sensing property of the tactile sensor after being embedded in the CFAS shows similar properties compared to the tactile sensor without CFAS (Fig. 3f). Therefore, embedding the tactile sensor with CFAS can neutralize the effect of structural obstruction during the wound-healing process, and this leads to accelerated skin regeneration while maintaining the pressure-sensing property of the tactile sensor. Consequently, the CFAS-embedded tactile sensor can serve as a substitute for permanently damaged tactile function while simultaneously accelerating the skin regeneration. Additional performance details of the CFAS-embedded tactile sensor can be found in Supplementary Fig. 17. The sensor is capable of measuring from a minimum sensing pressure of 60 Pa up to a saturation point of 0.1 MPa. The sensor also exhibits some level of

hysteresis, which is from a result of the force transmission delay due to the viscous characteristics of CFAS. In addition, structures other than the cuboid were investigated, as shown in Supplementary Fig. 18. As theorized earlier, since the height of the structure is the most critical factor for the bending effect, there was not a significant difference in the sensor's performance between two structures which has the same height. Reepithelialization and collagen deposition at day 21 were observed by hematoxylin and eosin (H&E) and Masson's tri-chrome (MT) staining (Fig. 3g). Through H&E staining, dermis regeneration (lower layer) and reepithelialization (upper layer) in repaired tissues were observed. The groups treated with CFAS exhibited a significantly full-thickness skin regeneration of the dermis and epidermis, and it was clearly divided into two layers. However, in the tactile sensor and control groups, the epidermis was not completely regenerated and newly formed tissues were heterogeneous with surrounding native tissues in histologic morphologies. MT staining was performed to observe the collagen deposition and remodeling in the regenerated skins. On day 21 of CFAS-treated groups, intensive collagen deposition and typically dense collagen fibers were observed under the epidermal layers. Conversely, sparse collagen fibers were identified in the tactile sensor group, and incomplete collagen deposition was observed in the control group. Overall, these histology staining results demonstrate that the wound-healing effect of the CFAS group was enhanced compared to groups without CFAS.

Furthermore, the wound-healing effects of the CFAS were confirmed via immunofluorescence (IF) staining (Fig. 3h) and quantitative analysis of IF staining images (Fig. 3i–k). For confirming the maturation of regenerated skin tissues, the expression of loricrin which is the epidermal protein was examined in wound sites. From the results, in CFAS and CFAS@Tactile sensor groups, loricrin was significantly highly expressed along the epidermal layer, but in tactile sensor and control groups, loricrin was sparsely expressed (Fig. 3h, i). Furthermore, to confirm the polarization of type 2 macrophages, which are associated with anti-inflammation and tissue remodeling in the wound-healing process, the expression of CD68 and CD206 was examined[54]. In Fig. 3h, j, CD68 and CD206 which are the markers of total macrophages (M0) and type 2 macrophages (M2), respectively, were expressed slightly in all groups, but the expression ratio of M2 to M0 showed a significant difference between the CFAS-treated groups and the tactile sensor group (Fig. 3j). In addition, apoptotic cells stained by caspase-3 were rarely expressed and not significantly different in all groups except the tactile sensor group indicates that the CFAS and CFAS@-Tactile sensor groups were nontoxic and biocompatible to surrounding tissues (Fig. 3k). Therefore, it is considered that the incorporation of CFAS promotes wound healing, skin remodeling, and accelerates skin regeneration.

## Biocompatibility of WTSA

When the materials are implanted into the body, excellent biocompatible materials cause minimized foreign body reactions and immune responses to surrounding living cells and tissues[55]. Therefore, in this study, we implemented optimized biocompatible strategies to enhance the compatibility of the fully implanted WTSA with the body, as well as ensure coverage of essential biological functions. In an in vitro, a cytotoxicity test was conducted on the components of the WTSA using live-dead staining and CCK-8 assay of NIH-3T3 cells (Fig. 4a). Compared to the control group, the similar morphology and density of cells visible in the live-dead staining images of the WTSA components confirm their non-toxicity to cells. Figure 4b presents the CCK-8 assay results, indicating that the cell viability rate of the WTSA components is over 80% higher than that of the control group; thus, the components of the WTAS are non-cytotoxic.

When an external material is implanted into the body, it can induce the formation of fibrotic collagen layers through a foreign body reaction. The thickening of the surrounding collagen layer, known as

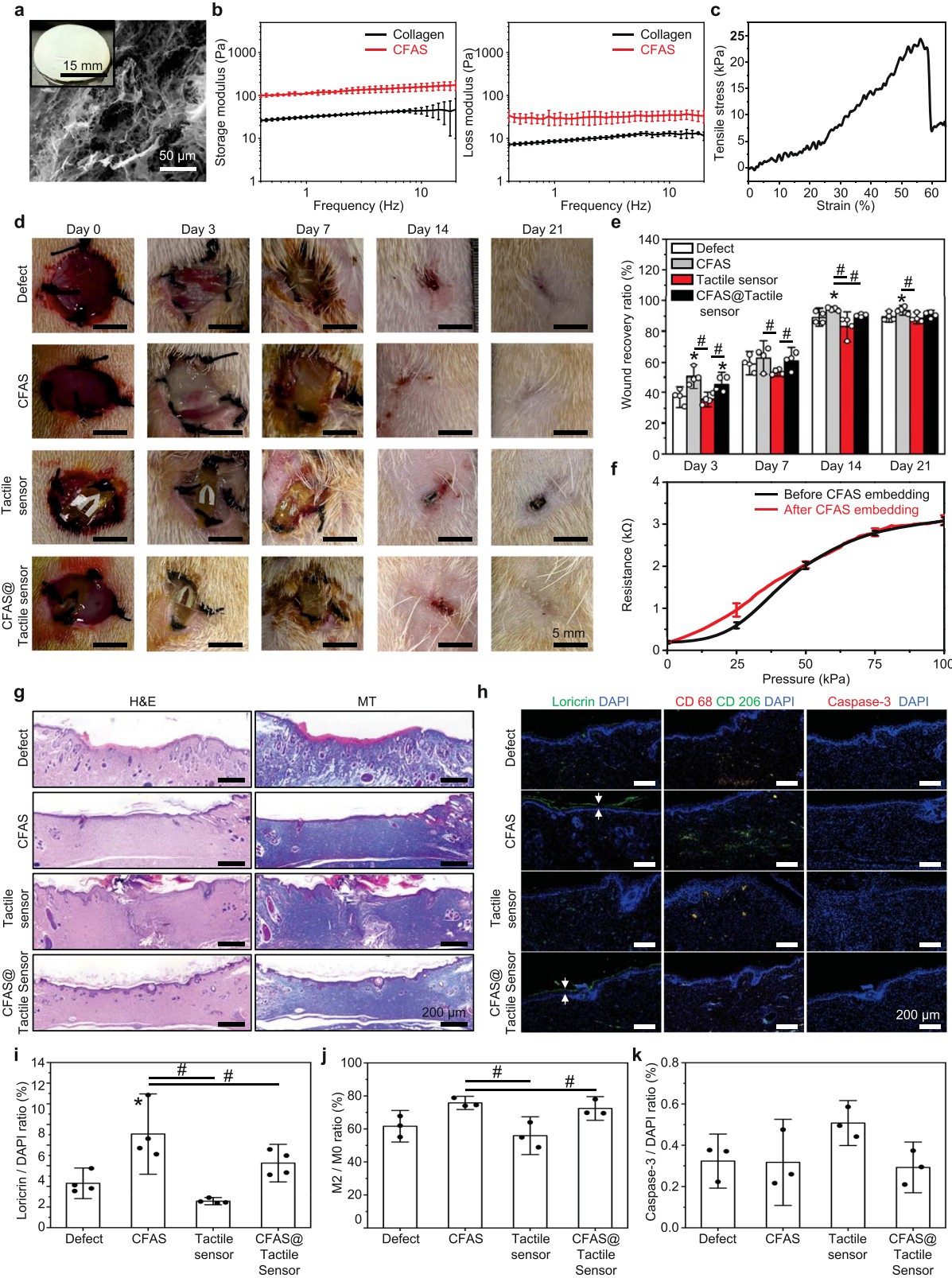

fibrosis or encapsulation, is a part of this inflammatory response. Encapsulation can sometimes lead to certain biological problems such as inflammation, impaired functionality, restricted movement, or mobility with implanted material. Further, these problems could cause discomfort, pain, and tissue damage near the site of encapsulation[55]. WPPFM, which is the part of WTSA, was fully implanted in the subcutaneous back of rats for 2 weeks, while a sham group (undergoing

the same surgical process without implantation) was used as the control. Collagen encapsulation by foreign body reaction and immune response was investigated by H&E staining, MT staining, and IF staining (Fig. 4c, d). Comparing the H&E staining images of the sham and WPPFM group on day 7, inflammatory mediators such as cytokines, chemokines, and growth factors were released in an acute inflammatory phase. These signaling molecules recruited immune cells,

**Fig. 3 | Characterization and wound-healing efficacy of the collagen (5 mg/mL) and fibrin (6.67 mg/mL) based artificial skin (CFAS). a** Representative scanning electron micrograph images and photograph (inset) of the CFAS. **b, c** Mechanical analysis results obtained for CFAS. **b** Storage (G', Pa) and loss modulus (G", Pa) versus frequency at isothermal temperature at 25 °C of hydrogel ($n = 3$). **c** Tensile test evaluated elasticity (Young's modulus, kPa) of CFAS. **d** Representative images of the wound on days 0, 3, 7, 14, and 21 following the various treatments, scale bar = 5 mm. **e** Quantitative analysis of wound recovery ratio on days 3, 7, 14, and 21 following treatment for all groups ($n = 4$). **f** Measuring resistance change of the tactile sensor embedded in CFAS ($n = 3$). **g** Representative H&E and MT-stained images of sectioned skin wound model tissues of 21 days after treatments, scale bar = 200 μm. **h** Immuno-fluorescent staining related with epidermis (loricrin; green), macrophages and anti-inflammation (CD68; red, CD206; green), and apoptosis cells (caspase-3; red) with DAPI (blue), scale bar = 200 μm. **i–k** Quantitative analysis of the activated cells ratio. **i** Loricrin to DAPI ratio ($n = 4$), **j** M2 to M0 ratio ($n = 3$), and **k** Caspase-3 to DAPI ratio ($n = 3$). All error bars are presented as mean ± SD. *P* values were analyzed by one-way ANOVA with Tukey's post hoc test. *$P < 0.05$, compared with that in the control group; #$P < 0.05$, compared with each group.

primarily neutrophils and macrophages, to the site of implantation. These recruited inflammatory cells can affect the fibroblast to induce a collagen layer, thus the collagen layers of the WPPFM group were slightly thicker than the sham to 200–300 μm on day 7 (Fig. 4c)[56,57]. Afterward, uncontrolled acute inflammation may become chronic, it can lead to continued collagen deposition and thickening of the capsule around the WPPFM. But in H&E staining images of the WPPFM group on day 14, it was confirmed that the collagen layer no longer thickened but became thinner to about 150–200 μm, similar to the sham group. This result suggested that no sustained inflammation occurred around WPPFM after the acute inflammatory phase. In addition, through the MT images on day 7, irregular collagen deposition was formed around WPPFM. However, on day 14, we observed collagen layer became stable and condensed similar to that of sham. Namely, WPPFM exhibited that it could minimize the result of foreign body reaction around the tissues and did not induce a chronic inflammatory reaction. Subsequently, to confirm the inflammation and toxicity of WTSA, immunofluorescence staining was performed using CD68, CD206, and caspase-3 antibodies to investigate the expression of M0 and M2 macrophages, as well as apoptotic cells (Fig. 4d). There was no significant difference observed between the WTSA and the sham group, indicating no immune response and noncytotoxicity.

Collagen encapsulation by foreign body reactions in the neural interface of nerve cuff electrodes blocks the electrical stimulation pathway between nerve and electrodes, resulting in the attenuation of signals transmitted to the nervous system. To suppress foreign body reactions on the neural interface electrodes for an effective electrical stimulation after long-term implantation, the hydrogel was coated on the bare neural interface electrodes of the WTAS. The hydrogel with a 3D scaffold has mechanical properties matched with nerve and hydrophilic property which reduce tissue damage as well as providing a hydrated environment. The hydrated environment (hydrogel) between the nerve and electrodes minimize the fibrosis formation, a process of the foreign body reaction, leading to its promising results for long-term in vivo stimulation[37,56]. In an in vivo, a rat sciatic nerve was wrapped by bare neural interface or hydrogel-coated neural interface of WTAS for 2 weeks and the sham group (following the surgical process) was used as control. Collagen layer formations, macrophage expression rate, and degree of apoptosis as a result of in vivo implantation were confirmed through histology analysis and IF staining (Fig. 4e–h). On day 14, the group coated with the hydrogel, as well as the sham group, exhibited significantly smaller fibrosis thickness compared to the bare neural interface group, as shown in the H&E staining image (Fig. 4e, f). Further, in the bare neural interface group of the MT staining, collagen fiber formation around the nerve was observed to be denser compared to the sham and hydrogel-coated groups. In the aspect of neural interface toxicity, there was no significant difference in caspase-3 expression between the sham and hydrogel-coated neural interface, but there were significant differences compared to the bare neural interface (Fig. 4g, h). As a result of the expression of caspase-3, the neural interface part of WTSA shows no toxicity which would not cause neuronal damage. The therapeutic effect on neuronal damage depending on the presence of the hydrogel can be demonstrated through Supplementary Fig. 19. We confirmed that a reduction in the expression of stained neural markers in the w/o hydrogel group at 14 days (Supplementary Fig. 19; white arrows)[58,59]. In addition, the effective nerve stimulation through w/ hydrogel-coated electrode after 14 days of implantation was confirmed, while the w/o hydrogel-coated electrode cannot deliver effective nerve stimulation after 14 days of implantation due to the comparably thick fibrosis layer between the nerve and the neural interface (Supplementary Fig. 20). The overall result of the experiment indicates that all components used in WTSA do not exhibit cytotoxicity. Furthermore, we did not observe severe foreign body reactions in vivo test around 2 weeks at the tissue-contacting part of WTSA, so confirmed WTSA had biocompatibility. In addition, the biocompatible hydrogel-coated neural interface electrodes of the WTSA exhibit a long-term and reliable operation as well as minimize foreign body reactions and immune responses.

## In vivo demonstration of WTSA with quantitative analysis of leg movements with respect to applied pressure

Figure 5 demonstrates the successful modulation of tactile signal that leads to a stimulation of sciatic nerve when pressure is applied to a damaged skin area by using WTSA implanted in the severe skin-damaged rat model. Supplementary Movie 1 presents the demonstration of the WTSA in rat model with and without wireless power transfer coil which indicates WTSA successfully functions under wireless power transferred condition. Figure 5a and Table 1 show the quantitative analysis of leg movements and EMG signal measured simultaneously[60] when different pressures are applied. To quantitatively analyze leg movement when external pressure is applied, three reference points (Knee joints, Ankle, Metatarsal head) were designated to determine the leg movement angles which were obtained by connecting the three reference points[31,61]. In addition to the three-point reference method, we further measured the EMG signal from the contracting leg muscle to enable a more precise analysis (Supplementary Fig. 21). When nerve stimulation exceeds the threshold value, applying low-frequency stimulation (< 10 Hz) results in repetitive muscle contraction and relaxation, causing the leg to shake. Conversely, higher frequencies (>10 Hz) lead to continuous muscle contractions, stiffening the leg and increasing the angle of movement[32] (Supplementary Movie 2). In addition, the resistance of tactile sensor which is embedded in CFAS saturates after 85 kPa resulting the saturation of the leg movement after 85 kPa (Supplementary Movie 3). As shown in Fig. 5a and Table 1, increasing applied pressure results in an increase of leg movement which is induced by increased output frequency of WPPFM. When low pressure is applied inducing the low-frequency output of WPPFM, leg movement of shaking is observed which is consistent with the theoretical analysis of nerve stimulation with low-frequency signal. When applied pressure is increased as stimulation frequency increases, leg movement angle and the frequency of EMG signal increased. The leg movement angle showed no difference when 84.5 kPa and 96.4 kPa pressure is applied while EMG signal showed 45.45 Hz and 45.75 Hz, respectively, indicating the EMG signals can precisely monitor the stimulation difference. Figure 5b and Supplementary Movie 4 show the successful wound healing and the leg movement response to the applied pressure

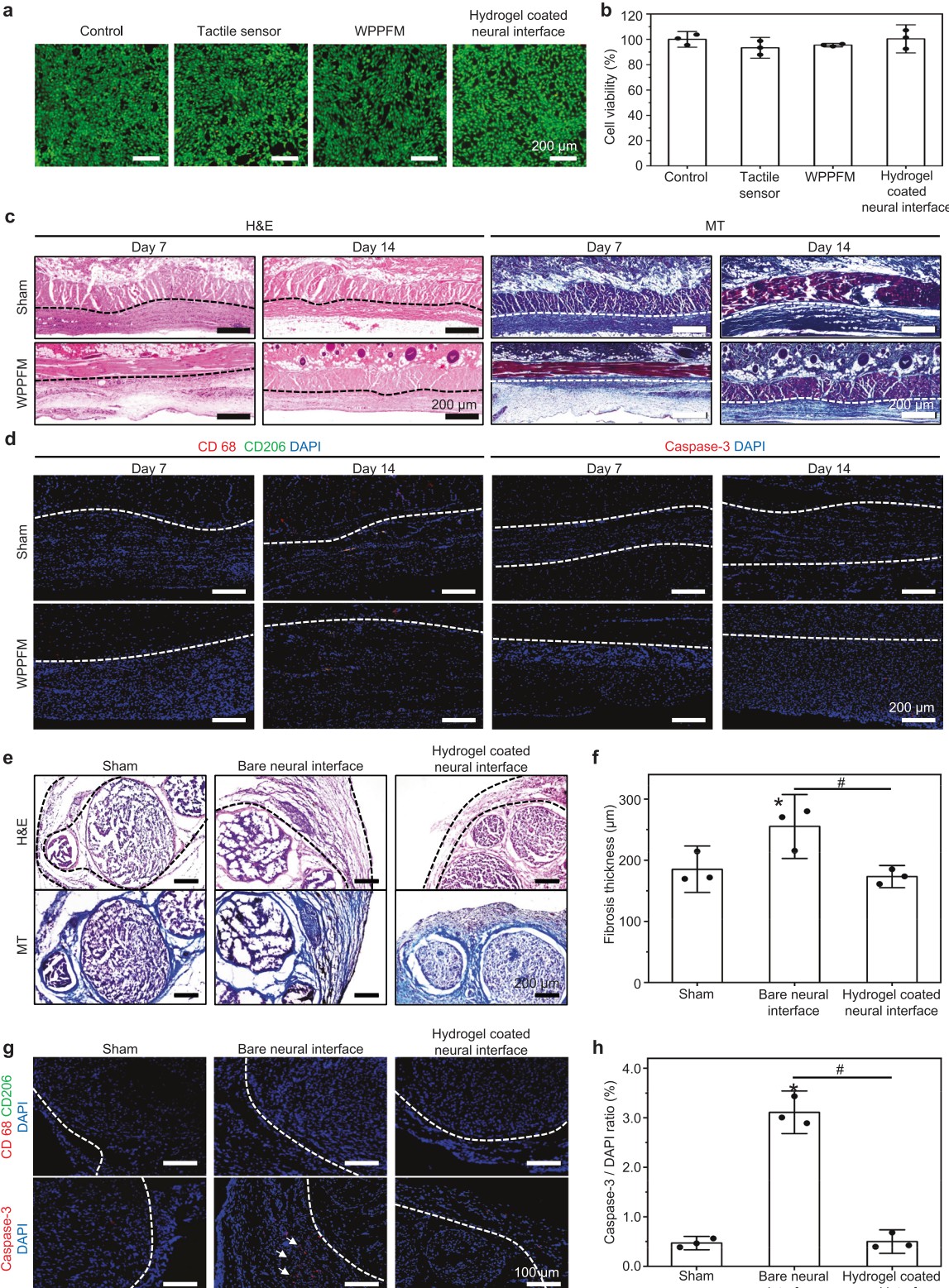

**Fig. 4 | Biocompatibility of the WTSA. a** Live-dead staining images of fibroblast cell (NIH-3T3) cultured with dissolution media with WTSA components, scale bar = 200 μm. **b** Cell viability rate of the WTSA components were evaluated using the CCK-8 assay (*n* = 3). **c** Representative H&E and MT-stained images of the skin layer where the WPPFM was implanted for 14 days, scale bar = 200 μm. **d** Immuno-fluorescent staining of macrophages and anti-inflammation (CD68; red, CD206; green) and apoptosis cell (caspase-3; red) images with DAPI (blue) of the cross-sectioned skin layer tissue where the WPPFM was implanted for 14 days, scale bar = 200 μm. **e** Representative H&E- and MT-stained images of cross-sectioned sciatic nerve which was wrapped with neural interface of WTSA for 14 days, scale

bar = 200 μm. The width of the black dot lines indicates the fibrosis thickness. **f** Quantitative analysis of fibrosis layer thickness (*n* = 3). **g** Immuno-fluorescent staining of macrophages and anti-inflammation (CD68; red, CD206; green) and apoptosis cells (caspase-3; red) images with DAPI (blue) of the cross-sectioned sciatic nerve wrapped with neuronal interface of WTSA for 14 days, scale bar = 100 μm. **h** Quantitative analysis of expressed caspase-3 to DAPI ratio (*n* = 3). All error bars are presented as mean ± SD. *P* values were analyzed by one-way ANOVA with Tukey's post hoc test. *$P < 0.05$, compared with that in the control group; #$P < 0.05$, compared with each group.

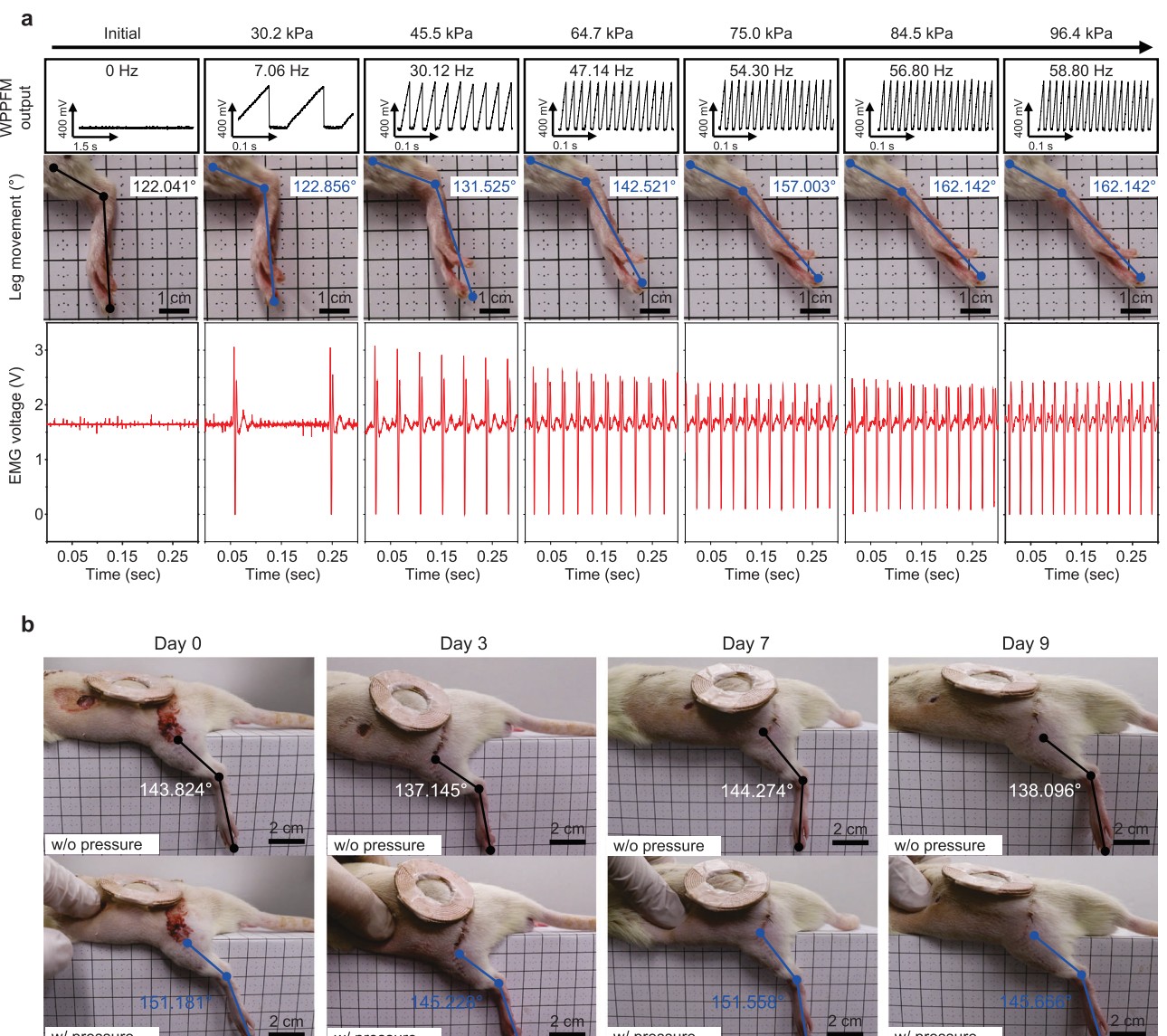

**Fig. 5 | In vivo demonstration of implanted WTSA and precise analysis of leg movements according to the externally applied pressure. a** Quantitative analysis of leg movement angle (blue line) and electromyogram signal (red) monitoring while different pressure is applied (Initial, 30.2, 45.5, 64.7, 75.0, 84.5, and 96.4 kPa). Inset black box represents the output signal from WPPFM, and the inset blue box represents the leg movement angle under an applied pressure. **b** In vivo demonstration of leg movements under an applied pressure from day 0 to day 9 (top column shows the initial angle of the leg; black line, bottom column shows the leg movement when pressure is applied; blue line).

**Table 1 | Quantitative analysis of leg movements according to different applied pressures**

| Applied pressure (kPa) | 0 | 15.4 | 30.2 | 45.5 | 64.7 | 75 | 84.5 | 96.4 |
|---|---|---|---|---|---|---|---|---|
| Nerve stimulation frequency (Hz) | 0 | 0.53 | 7.06 | 30.12 | 47.14 | 54.3 | 56.8 | 58.8 |
| Leg movement | 122.041 | 122.041 | 122.856 | 131.525 | 142.521 | 157.003 | 162.142 | 162.142 |
| Angle difference | 0 | 0 | 0.815 | 9.484 | 20.48 | 34.962 | 40.101 | 40.101 |
| EMG signal frequency | 0 | 0 | 22.01 | 24.65 | 35.17 | 44.03 | 45.45 | 45.75 |

over a 9-day period of WTSA implantation. When the damaged skin area was pressed with a finger at a similar pressure, the change in leg angle was 7.357°, 8.083°, 7.284°, and 7.57° on day 0, day 3, day 7, and day 9, respectively. These in vivo results indicate that the WTSA remains well encapsulated for operation without notable degradation, while fibrosis is effectively minimized by the hydrogel coated on the neural interface electrodes. As a result, an effective sciatic nerve stimulation is obtained via modulation of the tactile signals in the form of sawtooth pulse signals.

## Discussion

We present a new approach for replacing permanently damaged tactile sensory and accelerating the regeneration of severely damaged skin with a single substrate WTSA. The collagen- and fibrin-based WTSA accelerates skin regeneration by providing microenvironments that mimic the biological and mechanical properties of the native skin. The intensity of the applied pressure is detected by a tactile sensor embedded in the artificial skin and converted to a sawtooth pulse signal with different frequencies through WPPFM to stimulate the

sciatic nerve. With a hydrogel-coated neural interface, the WTSA is nontoxic and minimizes foreign body reactions, which are essential to obtain an implantable device with biocompatibility and effective nerve stimulation properties. Our single substrate integrated implantable WTSA device represents a new paradigm for replacing the tactile sensory function and simultaneously regenerating full-thickness skin to replace a severely damaged skin area. We demonstrated an in vivo experiment by implanting the WTSA in severely skin-damaged rat models and precisely analyzed the leg movements by measuring the EMG signals and leg movement angle under different intensities of external pressures applied to the defected skin. Furthermore, the implanted WTSA operated stably for over 9 days as well as effectively promoted the wound-healing process. These advancements highlight the potential application of the WTSA in basic as well as clinical research in biomedical and associated fields. Also, by developing such a device into a multi-channel tactile sensing and converting units, it can provide fundamental treatment for severe skin damage over a large area.

## Methods

### Fabrication of wireless-powered tactile sensory system

PDMS (15:1 mixing ratio of base and curing agent) was spin-coated on a glass substrate and cured at 70 °C for 30 min and at 110 °C for 2 h. Next, a 25-μm-thick polyimide film (Hanarotr) was cleaned sequentially with acetone, isopropanol alcohol (IPA), and distilled water (DI) and subsequently laminated on the PDMS-coated glass substrate. Then, Cr/Au (5 nm/300 nm) was thermally evaporated (KVE-T2000, Korea Vacuum Tech) and patterned with a photoresist (AZ-5214E, MicroChemicals). After wet etching Cr/Au, the photoresist was removed by cleaning sequentially with acetone, IPA, and DI water. To form the insulating layer, gentle $O_2$ plasma treatment ($O_2$ 50 SCCM, 100 W, 30 s) with a reactive ion etcher (RIE, Q190620-M01, Young Hi-tech) is proceeded prior to spin coating liquid polyimide (Sigma-Aldrich) at 3000 rpm. The spin-coated polyimide layer was cured at 110, 150, and 210 °C for 10 min, 30 min, and 2 h, respectively. To fill the micropores of spin-coated polyimide, 50 nm of $SiO_2$ was sputtered. AZ4620 was patterned on the insulating layer to form the vias and dry etched with an RIE ($CF_4$, 20 SCCM, 150 W, 120 s and $O_2$ 200 SCCM, 200 W, 500 s). Ti/Cu (5 nm/600 nm) was thermally evaporated and patterned as the top metal layer.

Next, we fabricated the tactile sensor. First, a thin layer of ALP (fatty acid-modified acrylic polyester diluted with 50% of methyl ethyl ketone; MCNET Co., South Korea) was blade-coated to partially cover the exposed electrode. The device was then placed on a hotplate at 90 °C to evaporate the solvent and cured by exposing to UV (MCNET Co., South Korea) light for 15 min. After curing, a droplet of polyurethane acrylate (PUA) (CFM110S; MCNET Co., South Korea) was dropped onto the ALP and covered by a hydrophobically surface-treated glass for the rolling process. During the rolling process, the PUA gradually covered half of the exposed electrode and became thinner. The device was then placed in the UV chamber (MCNET Co., South Korea) for 1 min to cure the PUA, and the glass was removed. To define the sensor shape, a shadow mask was aligned with the electrode, and a 20-nm-thick layer of Pt was deposited by sputtering (Q300TD, Quorum Inc.). To generate microcracks, the sensor part was bent and rolled on a rod with a diameter of 1 mm. To develop a cuboid structure at the front, PUA (CFM110S; MCNET Co., South Korea) was filled between the polyimide film and a film photomask, separated by a distance of 500 μm using spacers on the sides, with a quadrangle shape. The photomask side was placed on top and semicured by exposing to UV light for 1 min. The obtained cuboid structure was gently washed with IPA to remove the uncured PUA and then stored separately in a Petri dish and fully UV-cured for 6 h (MCNET Co., South Korea). To assemble the tactile sensor, ALP was applied on the

structure, which was then attached at the front of the tactile sensor. The assembled device was UV-cured for 1 h until the structure was firmly attached to the sensor. The chip components for WPPFM were soldered using a low-temperature solder paste (SMDSTSFP, Chip Quik) and rinsed with IPA to remove the residual flux.

The exposed part of the Au electrodes was electroplated using the Transene sulfite gold (TSG-250, Transene) solution (1:5; Transene sulfite gold:DI water) and VersaSTAT3 instrument (Princeton Applied Research, USA) to improve its electrical properties. A three-electrode setup was developed using Ag/AgCl as the reference electrode, a Pt coil as the counter electrode, and the Au electrodes as the working electrodes. The Au-black electroplating process was conducted by applying a voltage of −0.92 V for 1 min.

After delaminating the device from the glass substrate, a 50-nm-thick layer of $SiO_2$ was sputtered on the back side of the device for oxide bonding. On the Parylene-C-coated glass, PDMS (7:1 mixing ratio of base and curing agent) was spin-coated at 700 rpm and cured at 80 °C for 24 h. The cured PDMS layer was then UV-ozone treated for 10 min, and the device was laminated on the UV-treated PDMS layer, which formed oxide bonds after curing at 110 °C for 30 min. Additional PDMS was spin-coated on the device and cured at 80 °C for 24 h. The device was relaminated on the glass substrate for laser cutting. Then, a 2 μm thickness of Parylene C was deposited, followed by deposition of $Al_2O_3$ (50 nm) on both sides of the device by atomic layer deposition as the final encapsulation layer. Finally, the hydrogel was coated on the electrode.

### Preparation of a hydrogel for the neural interface electrodes

Gelatin (Sigma-Aldrich, USA) and genipin (Wako Chemicals, USA) were dissolved in 60 v/v% ethanol (in deionized water). First, 60% of gelatin was dissolved in deionized water (8 w/v% versus deionized water), and then, genipin was poured into the gelatin solution (2 w/w% versus gelatin). The mixed solution was crosslinked on a hotplate and then poured onto the Au-blacked Au electrodes. After curing the hydrogel for 2 days (23 °C, humidity: ~30%), the hydrogel was swelled in PBS at 37 °C for an additional 2 days before its use[37].

### Characterization of the hydrogel

A cylindrical-shaped hydrogel (diameter: 2 cm, height: ~1 cm) was prepared, whose compressive stress–strain curve was acquired using a tensile testing machine (INSTRON, USA). Then, the compressive modulus of the hydrogel was calculated as the slope of the stress–strain curve in the region with a strain value of up to 15%. To calculate a conductivity of the hydrogel, bulk hydrogel (Diameter: 9 cm, Height: ~1.5 cm) was prepared and Nyquist plots were acquired by a VersaSTAT3 (Princeton Applied Research, USA) with two stainless plate electrodes. The conductivity equation $\sigma = L/A$, × Rb, where L, Rb, and A denote the adjacent distance (2 mm), bulk resistance of the hydrogel obtained from the real axis ($Z_{re}$) of the Nyquist plot at 100 kHz, and area of the plate electrodes (1 cm × 1 cm), respectively, can be used to calculate the conductivities of the hydrogel and PBS[62,63].

### Generation of collagen and fibrin-based artificial skin hydrogel (CFAS)

Rat tail collagen I (354249, corning, USA) was purchased and diluted to 5 mg/mL as an optimized concentration. Collagen I, 10× MEM (Gibco), 1 N sodium hydroxide (Sigma-Aldrich, USA), DI water were prepared. Collagen solution was prepared by following the manufacturer's instructions. After then, Fibrinogen (6.67 mg/mL, Sigma-Aldrich) was dissolved with phosphate-buffered saline (PBS) in 37 °C water bath. To prepare the collagen/fibrin artificial skin hydrogel, an equal ratio (v/v) of solutions were mixed with thrombin (50 U/mL, Sigma-Aldrich) at a volume ratio of 100:1. Mixed solution was incubated at 37 °C for 30 min for gelation[42–44].

## Characterization of CFAS

Sample preparation for SEM (Inspect F50; FEI Company, Hillsboro, OR, USA), CFAS was fixed with 4% paraformaldehyde for 3 min, followed by dehydration in gradient ethanol solution (70%, 80%, 90%, and 100%). The dehydrated hydrogel was further dried by using a freezing dryer (ilshinBioBase, Korea) for 2 days. The dried sample was sputter-coated with gold (SPI-module sputter coater; SPI Supplies, West Chester, PA, USA) and observed with a scanning electron microscope (Tenoeo Volume Scope, FEI; Hillsboro, OR USA) at an acceleration voltage of 15 kV.

The viscoelastic behavior of a hydrogel was measured by a rheometer (MCR 102, Anton paar, Austria). The mold to measure the viscoelastic property of CFAS was manufactured with a diameter of 2 cm. The sample was loaded on the rheometer, and the test was conducted at different frequencies at an isothermal temperature of 25 °C.

Tensile test was conducted to measure a reaction to forces being applied in tension. The tensile properties of CFAS were investigated using an Instron machine (UTM, Instron 5966; Instron Corp, Norwood, MA, USA). A dumbbell-shaped (5 cm × 1.5 cm × 0.5 cm) hydrogel mold was manufactured, and three different samples were prepared for the test. The sample was fixed on Instron by super glue (Supplementary Fig. 16). Next, the samples were tested under a pressure of 1 kN at a speed of 20 mm/min.

## Wound defect formation and WTSA implantation

The Institutional Animal Care and Use Committee of KIST authorized all animal treatments and experimental procedures (KIST-IACUC-2022-016-2). Six-week-old 150–175 g SD male rats (DBL, Chungbuk, Korea) were purchased and cared for under specific pathogen-free conditions. In the animal experiment, all surgical procedures were performed under respiratory anesthesia. To induce a wound defect in rats, a round-shape full-thickness wound defect with a diameter of 10 mm was created on the back of each rat. The defect was locally treated using the prepared tactile sensor, CFAS, or CFAS with tactile sensor (CFAS@tactile sensor), and untreated wounds were used as the control group. Tegaderm™ (3 M Health Care, St. Paul, MN, USA), a commonly used dressing material, was applied to all groups for a duration of 7 days[64]. The wound-healing process was then observed for a period of up to 21 days after treatment initiation. Next, in vivo experiments were conducted to confirm the functional maintenance and biocompatibility of WTSA. Before implantation, the WTSA was subjected to cleaning and sterilization procedures. The WTSA was inserted into the subcutaneous layer after partially incising the left part of the rat's spine. The part of the tactile sensor was embedded with CFAS in the wound defect. Then, sciatic nerve was exposed, and the hydrogel neural interface of WTSA was wrapped around the nerve.

## H&E and MT staining for histological analysis

For the histological analysis, H&E and MT staining were conducted on cross-sectioned tissues. Related tissues with in vivo implantation were collected and fixed in 10% formalin for 2 days. Subsequently, fixed tissues were embedded in a frozen section solution (Leica, Germany), frozen using dry ice and LN2. Frozen blocks were sectioned in an optimal cutting temperature using microtome (CM3050 S; Leica) to 10–12 μm thickness. Sectioned tissues were dried on slides, stained with either H&E or MT using standard protocols, and observed under a light microscope (Olympus). The thicknesses of the fibrotic tissues were measured using the Image J software ($n = 5$ in each group).

## Immunofluorescence staining

For confirming immune response from tissues, immunofluorescence staining was conducted following standard protocols. Cross-sectioned slides were blocked with 4% bovine serum albumin solution (Sigma-Aldrich, USA), incubated with primary antibodies overnight, and followed standard washing protocol. Alex Fluor 594 anti-cluster of differentiation 68 (CD68 at 1:100; Santa Cruz Biotechnology), Alex Fluor 488 anti-cluster of differentiation 206 (CD206 at 1:100; Santa Cruz Biotechnology), anti-caspase-3 (1:100; Santa Cruz Biotechnology), and anti-loricrin (Loricrin at 1:100; Abcam) were used to investigate the distribution of macrophages, apoptosis, and epithelium in in vivo experiments. The CD68 + , CD 206 + , and caspase-3+ cells ratio was calculated as the percentage of the total cell area consisting of CD68- or caspase-3-area.

## Live/dead staining and CCK-8 assay

NIH-3T3s were purchased from Lonza (Walkersville, MD, USA) and cultured in cell culture flasks (Thermo fisher, USA) with Dulbecco's modified Eagle's medium (DMEM; Gibco, USA), supplemented with 10% (v/v) fetal bovine serum (FBS; Gibco) and 1% (v/v) penicillin-streptomycin (PS; Gibco), in a 5% carbon dioxide ($CO_2$) cell incubator at 37 °C. NIH-3T3 cells were seeded at a density of $1 \times 10^4$ cells per well in a 24-well plate and the culture medium was changed every second day. Components of WTSA were placed in fresh cell culture medium and extracted for 24 h with shaking in a 37 °C. Then, each extraction medium was diluted at a 1:1 (v/v) ratio of extraction medium to fresh culture medium. Before treating the diluted medium to cells, each diluted medium was filtered with a 0.45-μm filter. Treating the diluted medium to cells for 24 h. Thereafter, the medium was removed and treated with a diluted cell counting kit-8 (CCK-8) solution for 2 h to analyze the cell viability. The absorbance of the reaction solution was measured at a wavelength of 450 nm using a 96-well plate reader, and all samples were compared with a control to calculate the percentage of live cells.

The live/dead assay was performed using fluorescein diacetate (FDA, 1.5 mg/mL in acetone; Sigma-Aldrich) and ethidium bromide (EB, 1 mg/mL in PBS; Sigma-Aldrich). FDA end EB stain the cytoplasm of viable cells green and the nuclei of nonviable cells to red, respectively. The NIH-3T3 cells were then treated with the staining solution for 5 min at 37 °C. Following staining, the cells were washed three times with PBS and observed under a fluorescence microscope (Olympus).

## The procedure of measuring leg movement and EMG monitoring

When applying the pressure with finger, the precise value of applied pressure cannot be controlled. Therefore, converting the applied pressure value with the corresponding resistance value shown in Fig. 2f and by changing the resistance value, we monitored the EMG signal as well as leg movement simultaneously. To quantitatively analyze leg movement when external pressure is applied, three reference points (knee joints, ankle, metatarsal head) were designated to determine the leg movement angles which were obtained by connecting the three reference points[31,37,61].

Two methods are available for measuring EMG signals, including the needle-insertion-based method and conductive patch-based method. Needle-insertion EMG measurement method is invasive compared to the conductive patch-based method, it allows for more precise monitoring by directly recording the electrical signal from muscles. Therefore, we chose the needle-insertion EMG measurement method for more precise analysis of the leg movement according to applied pressure. Supplementary Fig. 21 shows the setup of needle-insertion EMG signal measurement by inserting the needles in leg muscle (soleus) which is responsible for leg contraction, with needle spacing of 1 cm and on the back side of the body for the reference electrode. Three electrodes were connected to the commercial EMG module (PSL-iEMG2, PhysioLab) which includes filters and amplifier for noise removal. Amplified and filtered signals are recorded by a voltage-based data acquisition system (DAQ) module (PXIe-6365, National Instruments) with the sampling frequency of 7000 Hz.

## Statistical analysis

All the data were presented as mean ± standard deviations and analyzed using a one-way analysis of variance (ANOVA) with a Tukey's significant difference post hoc test. *P* values of <0.05 were considered statistically significant. Statistical analyses were performed using Origin 2020 software.

## Reporting summary

Further information on research design is available in the Nature Portfolio Reporting Summary linked to this article.

# Data availability

All the data supporting the findings of this study are available within this paper and its Supplementary Information. Any additional information can be obtained from the corresponding author on request.

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

## Acknowledgements

This study was supported by the Nano Material Technology Development Program NRF-2018M3A7B4071106 (Y.J., K.J.Y., T.K., and H.Y.), by the Individual Basic Science & Engineering Research Program NRF-2021R1A2C2004634 (Y.J.), and by the Brain Research Program NRF-2019M3C7A1032076 (T.K.) through the National Research Foundation of Korea funded by the Ministry of Science and ICT of Korea; the National Research Foundation of Korea grant nos. NRF-2019R1A2C2086085 and NRF-2021R1A4A1031437 (K.J.Y.); the KIST Institutional Program Project no. 2E31603-22-140 (K.J.Y.); the Individual Basic Science & Engineering Research Program NRF-2021R1A2C3011450 (H.Y.) through the National Research Foundation of Korea funded by the Ministry of Science and ICT.

## Author contributions

Kyowon K., S.Y., C.J., J.J., H.Y., T.K., K.J.Y., and Y.J. conceived the idea and designed the research. Kyowon K., S.Y., C.J., J.J., Y.Y., J.-Y.J., Y.-J.K., S.L., and T.H.K. fabricated the integrated device. K.Y.K. and H.J. laser cut the integrated device. J.U.K., G.I.K., D.H.C., Kiho K., J.P., J.-H.H., B.P., Kyubeen K., S.J., J.Y., K.L., H.C., B.-W.M., and H.J.K. were participated in the discussion of the methods. K.B., D.C. did a pilot study about pressure signal conversion. H.Y., T.K., K.J.Y., and Y.J. supervised and directed this work. Kyowon K., S.Y., C.J., J.J., H.Y., T.K., K.J.Y., and Y.J. wrote the manuscript. All authors discussed and commented on the manuscript.

## Competing interests

The authors declare no competing interests.

## Additional information

[1]Department of Electrical and Electronic Engineering, Yonsei University, 50, Yonsei-ro, Seodaemun-gu, Seoul 03722, Korea. [2]Center for Biomaterials, Biomedical Research Institute, Korea Institute of Science and Technology (KIST), Seoul 02792, Korea. [3]Department of Applied Bioengineering, Graduate School of Convergence Science and Technology, Seoul National University, Seoul 08826, Korea. [4]School of Chemical Engineering, Sungkyunkwan University (SKKU), Suwon 16419, Republic of Korea. [5]Post-Silicon Semiconductor Institute, Korea Institute of Science and Technology, Seoul 02792, Republic of Korea. [6]Department of Biomedical Engineering, The University of Texas at Austin, Austin, TX 78712, USA. [7]School of Electrical and Electronic Engineering, YU-KIST Institute, Yonsei University, Seoul, Republic of Korea. [8]Department of Fusion Research and Collaboration, Biomedical Research Institute, Seoul National University Hospital, Seoul 03080, Republic of Korea. [9]Biomaterials Research Center, Biomedical Research Division, Korea Institute of Science and Technology, Seoul 02792, Republic of Korea. [10]Department of Mechanical Engineering, Seoul National University, 1 Gwanak-ro, Gwanak-gu, Seoul 08826, Republic of Korea. [11]Querrey Simpson Institute for Bioelectronics, Northwestern University, Evanston, IL 60208, USA. [12]Department of Electrical and Computer Engineering, University of Illinois Urbana-Champaign, Urbana, IL 61801, USA. [13]Department of Chemical Engineering, Stanford University, Stanford, CA 94305, USA. [14]Research Institute for Convergence Science, Seoul National University, Seoul 08826, Republic of Korea. [15]Department of Engineering Science and Mechanics, The Pennsylvania State University, University Park, PA 16802, USA. [16]KU-KIST Graduate School of Converging Science and Technology, Korea University, Seoul 02841, Republic of Korea. [17]Department of Materials Science and Engineering, YU-KIST Institute, Yonsei University, Seoul 03722, Republic of Korea. [18]These authors contributed equally: Kyowon Kang, Seongryeol Ye, Chanho Jeong, Jinmo Jeong. ✉e-mail: hjungyi@kist.re.kr; taeilkim@skku.edu; kijunyu@yonsei.ac.kr; winnie97@kist.re.kr

