## [Peer Review File · Nature Communications]

REVIEWER COMMENTS

Reviewer #1 (Remarks to the Author):

In this manuscript, Kang and coauthors reported a fully implantable, wirelessly powered tactile sensory system embedded artificial skin (WTSA) to restore permanently damaged tactile function and accelerate wound healing for regeneration of severely damaged skin. Overall, the authors presented an interesting concept of design, fabricated an integrated system, and demonstrated its in vivo functionality. However, this reviewer raised several questions, which should be addressed in a revised manuscript.

1. For the crack-based tactile sensor, the authors only showed the pressure-resistance curves. More evaluation should be included, such as the lower and upper limits of detection, resolution, and precision.
2. From Line 268 to 270, the authors described that the accommodation of the curved surface of the body was achieved by fine-tuning of passive element values. Can the authors comment in more detail on the relationship between the properties of curved surfaces and the passive element values?
3. From Line 406 to 408, the authors observed that collagen layers around WPPFM became thinner. Can the authors comment more on this observation? More discussion based on the mechanisms of acute and chronic inflammations should be added here to better support the claim on “minimized foreign body reaction”.
4. In Fig. 1c, the authors showed that effective electrical stimulation was enabled by the conductive hydrogel layer, which reduced foreign body reactions. However, this schematic was not directly proved since the authors only demonstrated that hydrogel reduced the fibrosis occurrence in Fig. 4. A control group of w/ and w/o hydrogel coating should also be tested for in vivo experiment to verify the therapeutic effect provided by the hydrogel’s conductive pathway.
5. In Supplementary Fig. 9, the authors showed the FEA result of pressure concentration during the bending of the sensor. However, the description in Line 253-254 that “only bends when broad pressure is applied and merely bends under localized pressure” is exactly the opposite of the FEA result in the black box, where the tactile sensor without the cuboid structure only bends when localized pressure is applied, and merely bends under broad pressure. Moreover, the pictures shown in Supplementary Fig. 9 missed the colored scale bar for pressure and the caption is ambiguous. A side-view schematic of the set-up should be provided, and the caption needs to be revised for easier understanding. Similarly, Figs. 2d-e did not have a clear indication of the location of the cuboid and the experimental set-up. Schematics with labels and arrows should be included.
6. In Supplementary Fig. 10 and Line 274-278, the authors described the PBS acceleration test based on Arrhenius law. However, it is not sufficient to use only two temperature points (70°C and 80°C) to extrapolate the failure time at 37°C. Since k and E_a are two unknowns in the equation, at least three conditions should be performed to validate the linearity of fitting, which provides a more accurate

prediction of failure time at 37°C. Also, a detailed calculation and extrapolation plot should be provided in the supplementary information.

7. In Fig. 5b, Supplementary Movie 3, and Line 471, the authors pressed the damaged skin with a finger at a similar pressure. What is the pressure here? It is observed from the movie that the pressure applied on Day 9 seems to be larger than that on Day 1. The pressure should be controlled in the same way as that in Fig. 5a by converting the applied pressure value with the corresponding resistance value shown in Fig. 2f, as mentioned in Line 670 (but should be 2f instead of 2e).

8. In Supplementary Figure 11, the resistance of hydrogel was calculated at 100 kHz (Line 289-290, Line 570), which is a frequency much higher than the one for nerve stimulation. The conductivity should be calculated with resistance at a lower frequency (e.g. 1 kHz). A Bode plot should also be presented as well. In addition, the term “conductive hydrogel” is inaccurate if it only shows “55% of the conductivity level in PBS”. It’s also not convincing that such low conductivity would contribute significantly to the effectiveness of electrical stimulation.

Minor issues:

9. In Supplementary Fig. 9, the authors showed that sensitivity to broad pressure could be improved by introducing a cuboid structure. Did the authors compare this cuboid structure with other structures with different geometries or arrangements such as pyramidal arrays?

10. In Line 281, the difference between an Au-blackened Au electrode and Au electrode should be explained, along with a background introduction of why an Au-blackened Au electrode has better electrical performance.

11. In Line 428, the figure number is missing in the parenthesis.

Reviewer #2 (Remarks to the Author):

The paper “Bionic Artificial Skin with a Fully Implantable Wireless Tactile Sensory System for Wound Healing and Restoring Skin Tactile Function” demonstrates a novel approach to replace permanently damaged tactile sensory and accelerate regeneration of severely damaged skin. The complete device in this research with good biocompatibility and effective nerve stimulation properties is impressive, so this review recommend to publish the article in the journal, with some minor revisions below.

Comment 1. There is no fully completed device figure with CFAS in the main figures. It would be better if you replace Fig. 2a with WTSA of Supplementary Fig. 4.

Comment 2. There is no mention of whether the experiments in Fig. 2d,e are the result of the tactile sensor being inside the CFAS. If they are, it would be better to indicate the 'CFAS' in the text or in the figure.

Comment 3. I think the correlation between the dimension and effect of the cuboid structure is lacking. Is there no correlation between length or width?

Comment 4. In the fig. 2d,e, the results seem to be explained well, but the explanation of the principle seems to be lacking. If you can add an image of the principle, it will be helpful to understand.

Comment 5. Please add a scale bar to Supplementary Fig. 9b.

Comment 6. Regarding the experiment in Supp. Fig. 9b, the $h=500$ $w=2$ case should have more force concentrated at the tip than the $h=0$ $w=2$ case, which would fit your logic, but isn't the data the opposite?

Reviewer #3 (Remarks to the Author):

The manuscript presents a sophisticated wireless-powered tactile sensory system embedded in artificial skin (WTSA). This innovation aims to restore tactile function and accelerate wound healing in severe skin damage caused by burns, accidents, or other traumas. The tactile sensing involves perceiving external pressures and translating them into frequency-based signals. Subsequently, these signals trigger nerve stimulation, effectively emulating tactile sensations and inducing leg movements. The wound healing effect is designed by incorporating collagen and fibrin-based artificial skin (CFAS). The entire system is implantable with flexible-PCB-based embedded system capable of wireless power transfer.

The concept of a fully implantable wireless sensory system appears to be novel. Integrating distinct elements to reestablish tactile sensation and expedite wound healing may offer a unique avenue to address skin damage. It is less clear though what the key fundamental innovation here is beyond the sophisticated system integration, which certainly is a solid achievement. In addition, this reviewer also has the following concerns regarding the significance of this work and its detailed research approach.

1. To add into the significance of the work, the authors are recommended to discuss how many patients are affected by severe skin damage, which also leads to permanent tactile sensory impairment.

2. The tactile sensing has just one sensor, with no spatial resolution. This is quite different from normal skin, where there is a high density of mechanoreceptors. It is hard to imagine how the proposed WTSA can scale up to many sensors.

3. Neural stimulation is typically charge-balanced, while this paper uses a single phase, which can lead to safety concerns.

4. The nerve stimulation is applied to the efferent nerve that drives the leg movement, but the tactile sensing is done through the afferent nerve. It is unclear how the demonstration done in this work can be effective for the recovery of tactile sensing.

5. The wound healing process doesn't seem to be significantly affected by the CFAS. The claim of accelerating wound healing may be premature.

Reviewer #1

General Comment: In this manuscript, Kang and coauthors reported a fully implantable, wirelessly powered tactile sensory system embedded artificial skin (WTSA) to restore permanently damaged tactile function and accelerate wound healing for regeneration of severely damaged skin. Overall, the authors presented an interesting concept of design, fabricated an integrated system, and demonstrated its in vivo functionality. However, this reviewer raised several questions, which should be addressed in a revised manuscript.

Our response: We thank the reviewer for these positive comments. We have carefully addressed the issues by including in-depth details and recused the manuscript accordingly.

Comment #1: For the crack-based tactile sensor, the authors only showed the pressure-resistance curves. More evaluation should be included, such as the lower and upper limits of detection, resolution, and precision.

Our response: Thank you for your comment. To demonstrate the fundamental performance of the sensor overall, we have included the relevant information in Supplementary Figure 17 and provided additional explanations in the manuscript.

Our modification to the manuscript 1:

Page 9, line 359-362 “Additional performance details of the CFAS-embedded sensor can be found in Supplementary Fig. 17. The sensor is capable of measuring from a minimum sensing pressure of 60 Pa up to a saturation point of 100 kPa. The sensor also exhibits some level of hysteresis, which is from a result of the force transmission delay due to the viscous characteristics of CFAS.”

Our modification to the manuscript 2:

Supplementary Fig. 17 | Basic tactile sensor characteristics embedded in the CFAS. The minimum pressure detection is 60 Pa (a), and due to the viscous environment of CFAS, there is some level of hysteresis (b).

Comment #2: From Line 268 to 270, the authors described that the accommodation of the curved surface of the body was achieved by fine-tuning of passive element values. Can the authors comment in more detail on the relationship between the properties of curved surfaces and the passive element values?

Our response: We thank the reviewer for pointing out this. Given the relatively low Young's modulus of the organs and tissues, implantable devices must exhibit flexibility to mitigate potential damage to surrounding tissues resulting from mechanical mismatch (rigidity). Due to the curved surface property of the tissue, slight bending after implanting the flexible device is inevitable. For the crack-based tactile sensor integrated within the device that we proposed, externally applied pressure induces the bending to the tactile sensor increasing the resistance, which subsequently translates into a voltage change across the voltage divider. The ring oscillator modulates the voltage change into a frequency-based sawtooth output signal. In the ring oscillator devised in this study, an operational minimum threshold voltage exists for voltage-to-sawtooth signal conversion. Accordingly, as shown in Supplementary Fig. 5b, precise adjustment of the fixed resistance value ($150\ \Omega$) within the voltage divider negates signal modulation induced by resistance changes due to the tissue curvature-related bending. To help understand this part, we have added more data points to Figure. 2f (red box) that the frequency output of the WPPFM stays zero until 16 kPa. Also, we have modified the manuscript for better readership.

Our modification to the manuscript:

To clarify this point, we have modified the main figure and following sentences in the main text.

Figure. 2f

Page 6, line 229-232, “The regulated voltage supplies lowered constant voltage (1.2 V) for device operation by voltage divider circuit which is composed of fixed resistor (R , 150 Ω) and resistance-based tactile sensor ($\Delta R_{\text{pressure}}$) resulting the resistance change of tactile sensor can be converted into voltage change ($1.2 \times \frac{R}{R + \Delta R_{\text{pressure}}}$).”

Page 7, line 271-274, “To accommodate the curved surface characteristics of the body, fine-tuning the fixed resistance value of voltage divider (150 Ω) allows the avoidance of frequency output when the tactile sensor is attached on curved surfaces and slightly bent without external pressure.”

Comment #3: From Line 406 to 408, the authors observed that collagen layers around WPPFM became thinner. Can the authors comment more on this observation? More discussion on the mechanisms of acute and chronic inflammations should be added here to better support the claim on “minimized foreign body reaction”.

Our response: We thank the reviewer for this careful comment. Previously, in the manuscript, as the reviewer said, there seemed to be a lack of explanation about the thinning of the collagen layer and the correlation with foreign body reaction. Therefore, on day 7 and day 14 after WPPFM implantation, observation of the collagen layer thickness was more discussed based on the mechanisms of acute and chronic inflammations. We have revised the manuscript to provide a more explanation about our study.

Our modification to the manuscript:

To clarify this point, we have modified the manuscript.

Page 10, line 411-412 “The thickening of the surrounding collagen layer, known as fibrosis or encapsulation, is a part of this inflammatory response.”

Page 10, line 419-430 “Comparing the H&E staining images of the sham and WPPFM group on day 7, in an acute inflammatory phase, inflammatory mediators such as cytokines, chemokines, and growth factors were released. These signaling molecules recruited immune cells, primarily neutrophils and macrophages, to the site of implantation. These recruited inflammatory cells can affect the fibroblast to induce a collagen layer, thus the collagen layers of the WPPFM group were slightly thicker than the sham to 200-300 μm on day 7 (Fig. 4c)^{56, 57}. Afterwards, uncontrolled acute inflammation may become chronic, it can lead to continued collagen deposition and thickening of the capsule around the WPPFM. But in H&E staining images of the WPPFM group on day 14, it was confirmed that the collagen layer no longer thickened but became thinner to about 150-200 μm , similar to the sham group. This result suggested that no sustained inflammation occurred around WPPFM after the acute inflammatory phase.”

Page 10, line 431-432 “However, on day 14, we observed collagen layer became stable and condensed similar to that of sham.”

Added references

56. Capuani S, Malgir G, Chua CYX, Grattoni A. Advanced strategies to thwart foreign body response to implantable devices. *Bioengineering & Translational Medicine* 2022, 7(3): e10300.
57. Pfisterer K, Shaw LE, Symmank D, Weninger W. The Extracellular Matrix in Skin Inflammation and Infection. *Frontiers in Cell and Developmental Biology* 2021, 9.

Comment #4: In Fig. 1c, the authors showed that effective electrical stimulation was enabled by the conductive hydrogel layer, which reduced foreign body reactions. However, this schematic was not directly proved since the authors only demonstrated that hydrogel reduced the fibrosis occurrence in Fig. 4. A control group of w/ and w/o hydrogel coating should also be tested for in vivo experiment to verify the therapeutic effect provided by the hydrogel's conductive pathway.

Our response: We appreciate the reviewer's comment. We acknowledge that our description of the hydrogel layer in the original Fig. 1c was slightly misleading. In the original Fig. 1c, we intended to illustrate that the hydrogel-coated electrode (w/ hydrogel) resulted in more effective stimulation due to the reduced fibrosis than the bare electrode (w/o hydrogel). The fibrotic encapsulation resulting from the foreign body reaction (FBS) is known to limit the function of the implants by increasing the impedance between tissue and implants^{R1, 36, R2, 37, R3}. For a clearer illustration, we modified Fig. 1c by deleting relevant phrases related to the electrical stimulation and the fibrosis.

In this study, the anti-foreign body reaction function of the hydrogel layer has been validated by results of in vivo experiments, where the hydrogel-coated neural interface group (w/ hydrogel) showed not only lower fibrosis thickness but also significantly less caspase expression compared with that of the bare neural interface group (w/o hydrogel) (Figure 4e-h). The soft hydrogel with the mechanical modulus being comparable with the sciatic nerve can

prevent the direct contact of the nerve with the stiff nerve electrode and also can provide hydrated environment to the nerve. However, we agree with the reviewer that further in vivo experiments are necessary to verify therapeutic effect of the hydrogel. To address the reviewer's comment, we performed additional in vivo experiments. Specifically, we examined the nerve damage with immunofluorescent staining (IF) and compared the degree of nerve damage and effect of electrical nerve stimulation of the electrodes w/ and w/o hydrogel coatings after implantation for 14 days.

The nerve damage was examined by IF staining of cross sectioned sciatic nerve with β -tubulin antibody as a neurofilament marker and S-100 antibody as a Schwann cell-specific marker^{58, 59}. As a result, we confirmed that a reduction in the expression of beta-tubulin and S-100 was observed in the w/o hydrogel group at 14 days, indicating some degree of nerve damage (white arrow). In contrast, the w/ hydrogel group showed a pattern of S-100 and beta-tubulin expression similar to that of the sham group, suggesting preserved expression levels. Additionally, we confirmed that the effective nerve stimulation can be delivered to the nerve through w/ hydrogel coated electrode after 14 days of implantation, while the w/o hydrogel coated electrode cannot deliver effective nerve stimulation after 14 days implantation due to the thick fibrosis layer between the nerve and the neural interface.

Supplementary Fig. 19 | Immunofluorescence staining of cross sectioned sciatic nerve in w/ and w/o hydrogel coated groups. (Red: S-100 (Schwann cell), Green: β -tubulin (neuron), Blue: DAPI (nucleus))

Supplementary Fig. 20 | Effectiveness of hydrogel coating on the neural interface for electrical stimulation after 14 days of implantation.

Our modification to the manuscript: We have revised the figure and manuscript to provide a clearer explanation of our study. Additionally, we have added an additional result of the experiments to support our claims.

Page 11, line 460-468 “The therapeutic effect on neuronal damage depending on the presence of the hydrogel can be demonstrated through Supplementary Fig. 19. We confirmed that a reduction in the expression of stained neural markers in the w/o hydrogel group at 14 days, indicating some degree of nerve damage (Supplementary Fig. 19; White arrows)^{58, 59}. Additionally, the effective nerve stimulation through w/ hydrogel coated electrode after 14 days of implantation was confirmed, while the w/o hydrogel coated electrode cannot deliver effective nerve stimulation after 14 days implantation due to the comparably thick fibrosis layer between the nerve and the neural interface (Supplementary Fig. 20).”

Fig. 1c

Supplementary Fig. 19 | Immunofluorescence staining of cross sectioned sciatic nerve in w/ and w/o hydrogel coated groups. (Red: S-100 (Schwann cell), Green: β -tubulin (neuron), Blue: DAPI (nucleus))

Supplementary Fig. 20 | Effectiveness of hydrogel coating on the neural interface for electrical stimulation after 14 days of implantation.

Added references

- R1. Farah S, Doloff JC, Müller P, Sadraei A, Han HJ, Olafson K, *et al.* Long-term implant fibrosis prevention in rodents and non-human primates using crystallized drug formulations. *Nature Materials* 2019, **18**(8): 892-904.
36. Zhang D, Chen Q, Shi C, Chen M, Ma K, Wan J, *et al.* Dealing with the Foreign-Body Response to Implanted Biomaterials: Strategies and Applications of New Materials. *Advanced Functional Materials* 2021, **31**(6): 2007226.
- R2. Meijs S, Fjorback M, Jensen C, Sørensen S, Rechendorff K, Rijkhoff NJM. Influence of fibrous encapsulation on electro-chemical properties of TiN electrodes. *Medical Engineering & Physics* 2016, **38**(5): 468-476.
37. Jeong J, Kim TH, Park S, Lee J, Chae U, Jeong J-Y, *et al.* Hybrid neural interfacing devices based on Au wires with nanogranular Au shell and hydrogel layer for anti-

- inflammatory and bi-directional neural communications. *Chemical Engineering Journal* 2023, **465**: 142966.
- R3. Goding JA, Gilmour AD, Aregueta-Robles UA, Hasan EA, Green RA. Living Bioelectronics: Strategies for Developing an Effective Long-Term Implant with Functional Neural Connections. *Advanced Functional Materials* 2018, **28**(12): 1702969.
58. Ronchi G, Fregnan F, Muratori L, Gambarotta G, Raimondo S. Morphological Methods to Evaluate Peripheral Nerve Fiber Regeneration: A Comprehensive Review. *International Journal of Molecular Sciences* 2023, **24**(3): 1818.
59. Ehmedah A, Nedeljkovic P, Dacic S, Repac J, Draskovic-Pavlovic B, Vučević D, *et al.* Effect of Vitamin B Complex Treatment on Macrophages to Schwann Cells Association during Neuroinflammation after Peripheral Nerve Injury. *Molecules* 2020, **25**(22): 5426.

Comment #5: In Supplementary Fig. 9, the authors showed the FEA result of pressure concentration during the bending of the sensor. However, the description in Line 253-254 that “only bends when broad pressure is applied and merely bends under localized pressure” is exactly the opposite of the FEA result in the black box, where the tactile sensor without the cuboid structure only bends when localized pressure is applied, and merely bends under broad pressure. Moreover, the pictures shown in Supplementary Fig. 9 missed the colored scale bar for pressure and the caption is ambiguous. A side-view schematic of the set-up should be provided, and the caption needs to be revised for easier understanding. Similarly, Figs. 2d-e did not have a clear indication of the location of the cuboid and the experimental set-up. Schematics with labels and arrows should be included.

Our response: We thank the reviewer for this careful review. The mentioned content has been appropriately revised. In addition, the colored scale bar and clearer captions have been added for Supplementary Fig. 9. For the side-view schematic, we intend to replace it with Supplementary Fig. 10. In this image, it illustrates how the sensor is embedded and how the wide-area rod applies pressure to the entire sensor with a cuboid structure.

Our modification to the manuscript:

Page 7, line 250-254 “Therefore, the tactile sensor with the cuboid structure (Supplementary Fig. 9; Blue box) effectively bends and shows almost identical stress distribution under localized and broadly applied pressure while tactile sensor without the cuboid structure (Supplementary Fig. 9; Black box) only bends when localized pressure is applied and merely bends under broad-area pressure.”

Supplementary Fig. 9 | **a**, Schematic illustration providing an overview of the experimental setup for the tactile sensor property test. w signifies the width of the rod, while h represents the height of the cuboid structure. **b**, FEM longitudinal strain results under different conditions. It compares sensors without and with a cuboid structure, denoted as $h = 0 \mu\text{m}$ (black boxes) and $h = 500 \mu\text{m}$ (blue boxes), respectively. The sensors were pressed by a rod with different diameters, namely $w = 2 \text{ mm}$ and $w = 20 \text{ mm}$. In the absence of a cuboid structure, it can be confirmed that even when pressure is clearly applied over a wide area, such as $w = 20 \text{ mm}$, no strain is formed on the sensor.

Supplementary Fig. 10

Supplementary Fig. 10 | A schematic image illustrating the theoretical operating mechanism of the cuboid structure. It displays images of the sensor's deformation with broadly applied pressure when a film-type sensor with a cuboid structure is embedded in soft material.

Comment #6: In Supplementary Fig. 10 and Line 274-278, the authors described the PBS acceleration test based on Arrhenius law. However, it is not sufficient to use only two temperature points ($70 \text{ }^\circ\text{C}$ and $80 \text{ }^\circ\text{C}$) to extrapolate the failure time at $37 \text{ }^\circ\text{C}$. Since k and E_a are two unknowns in the equation, at least three conditions should be performed to validate the linearity of fitting, which provides a more accurate prediction of failure time at $37 \text{ }^\circ\text{C}$. Also, a detailed calculation and extrapolation plot should be provided in the supplementary information.

Our response: We thank the reviewer for this careful review. As reviewer suggested, we have included the PBS acceleration encapsulation test data at $92 \text{ }^\circ\text{C}$ to validate the linearity of fitting,

which provides a more accurate prediction of failure time at 37 °C. The failure times at 70 °C, 80 °C, 92 °C are 248 hours, 92 hours, 29.84 hours respectively. The Arrhenius equation based on 70 °C and 80 °C is given by the formula: $lifetime (hour) = 1.533 \cdot 10^{-13} \cdot e^{\frac{99909.629}{8.314 \cdot T}}$. The estimated lifetime of our encapsulation at 92 °C based on the Arrhenius equation is 30 hours while the PBS acceleration test result showed 29.84 lifetime time at 92 °C which shows only 0.5% difference compared to the theoretical estimation. This result indicates that Arrhenius equation that we calculated can predict accurate failure time.

Our modification to the manuscript: We have added the accelerated Mg encapsulation test result at 92 °C to provide a more accurate prediction of failure time at 37 °C and the extrapolation plot including detailed calculation in the Supplementary Fig. 11 and Supplementary Note. 2. In addition, we modified the manuscript for better readability.

Page 7, Line 276-282 “The phosphate-buffered saline (PBS) acceleration test shows that Mg degrades after 248 h, 92 h and 29.8 h under 70 °C, 80 °C and 92 °C PBS solution, respectively, due to the failure of the encapsulation, indicating that the proposed multilayer encapsulation can survive for 429 days at a physiological temperature (37 °C), which can be calculated by the Arrhenius equation; $Lifetime = 1.533 \cdot 10^{-13} \cdot$

$e^{-\frac{99909.629}{8.314 \cdot T}}$, where 99909.629 is an activation energy, 8.314 is the gas constant, and T is temperature^{33, 34, 35} (Supplementary Fig. 11 and Supplementary Note. 2).”

Supplementary Fig. 11 | PBS acceleration test for multilayer encapsulation. Mg, which instantly reacts with water, was used to determine the lifetime of the multilayer encapsulation. At 70 °C PBS, 80 °C PBS and 92 °C PBS lifetimes of the multilayer encapsulation layer are 248 h, 92 h, 29.84 h respectively.

Supplementary Note 2

Calculating the lifetime of encapsulation is a crucial factor for implantable device. Since the body temperature of the animal is maintained at 37 °C, it is essential to assess the encapsulation lifetime at 37 °C. Therefore, we performed an accelerated PBS test using Mg which reacts immediately with water molecules. The lifetime of encapsulation layer is 248 hours, 92 hours and 29.84 hours at 70 °C, 80 °C and 92 °C respectively (Supplementary Fig. 11). The estimated lifetime of encapsulation layer can be calculated with the Arrhenius equation:

$$Lifetime = A \cdot e^{-\frac{E_a}{RT}} \quad (1)$$

Where A is constant, E_a is activation energy, R is gas constant and T is temperature. By substituting lifetime and temperature in equation (1), we get:

$$248 = A \cdot e^{-\frac{E_a}{8.314 \cdot 353.15}} \quad (2)$$

$$92 = A \cdot e^{-\frac{E_a}{8.314 \cdot 363.15}} \quad (3)$$

By calculating the equation (2) and (3), constant A is $1.533 \cdot 10^{-13}$ and activation energy (E_a) is 99909.629. We get the complete Arrhenius equation by substituting these parameters:

$$Lifetime = 1.533 \cdot 10^{-13} \cdot e^{-\frac{99909.629}{8.314 \cdot T}} \quad (4)$$

To confirm the linearity of calculated Arrhenius equation, estimated lifetime at 92 °C from equation (4) and measured lifetime at 92 °C were compared. The estimated lifetime at 92 °C is 30.07 hours, while measured lifetime at 92 °C is 29.84 hours which showed only 0.7% difference indicating that calculated Arrhenius equation can estimate the lifetime of encapsulation precisely. Therefore, the lifetime of encapsulation layer at 37 °C is 10,296 hours according to equation (4) which is equivalent to 429 days.

Added references

33. Song E, Li R, Jin X, Du H, Huang Y, Zhang J, *et al.* Ultrathin Trilayer Assemblies as Long-Lived Barriers against Water and Ion Penetration in Flexible Bioelectronic Systems. *ACS Nano* 2018, **12**(10): 10317-10326.
34. Fang H, Zhao J, Yu KJ, Song E, Farimani AB, Chiang C-H, *et al.* Ultrathin, transferred layers of thermally grown silicon dioxide as biofluid barriers for biointegrated flexible electronic systems. *Proceedings of the National Academy of Sciences* 2016, **113**(42): 11682-11687.
35. Fang H, Yu KJ, Gloschat C, Yang Z, Song E, Chiang C-H, *et al.* Capacitively coupled arrays of multiplexed flexible silicon transistors for long-term cardiac electrophysiology. *Nature Biomedical Engineering* 2017, **1**(3): 0038.

Comment #7: In Fig. 5b, Supplementary Movie 3, and Line 471, the authors pressed the damaged skin with a finger at a similar pressure. What is the pressure here? It is observed from the movie that the pressure applied on Day 9 seems to be larger than that on Day 1. The pressure should be controlled in the same way as that in Fig. 5a by converting the applied pressure value with the corresponding resistance value shown in Fig. 2f, as mentioned in Line 670 (but should be 2f instead of 2e).

Our response: We appreciate the reviewer's sharp comment. Supplementary Movie 4 (Original file name: Supplementary Movie 3) demonstrates the WTSA's successful conversion

of externally applied pressure into a frequency-based sawtooth signal, showing its ability to stimulate the nerve with respect to the different applied pressure after 9 days of implantation. We conducted an additional in vivo experiment that shows the resistance change of the tactile sensor while applying the pressure with pressing equipment (Instron) for quantitatively applying the pressure. Due to the characteristic of the tactile sensor in collagen and fibrin-based artificial skin (CFAS), the resistance of the tactile sensor maintains a certain value (3.1 k Ω) after threshold pressure of 85 kPa is reached (Supplementary Movie 3). Furthermore, the output frequency from the Wireless Powered Pressure-to-Frequency Modulation (WPPFM) circuit, responsible for converting the intensity of applied pressure (resistance) into frequency-based sawtooth signals, saturates after 85 kPa of applied pressure due to the saturated resistance. In Supplementary Movie 4, pressures ranging from 0 to 95 kPa was applied. It's important to note that once the threshold pressure of 85 kPa is reached, both the tactile sensor and WPPFM output saturate. Therefore, any additional pressure applied beyond 85 kPa does not exert a discernible impact on the strength of nerve stimulation.

Our modification to the manuscript:

We have added Supplementary Movie 3 and made modifications to the manuscript for better readability.

Page 12, Line 487-489 “In addition, the resistance of tactile sensor which is embedded in CFAS saturates after 85 kPa resulting the saturation of the leg movement after 85 kPa (Supplementary Movie. 3).”

Page 17, Line 702-704 “Therefore, converting the applied pressure value with corresponding resistance value shown in Fig. 2f and by changing the resistance value, we monitored the EMG signal as well as leg movement simultaneously.”

Comment #8: In Supplementary Figure 11, the resistance of hydrogel was calculated at 100 kHz (Line 289-290, Line 570), which is a frequency much higher than the one for nerve stimulation. The conductivity should be calculated with resistance at a lower frequency (e.g.,

1 kHz). A Bode plot should also be presented as well. In addition, the term “conductive hydrogel” is inaccurate if it only shows “55% of the conductivity level in PBS”. It’s also not convincing that such low conductivity would contribute significantly to the effectiveness of electrical stimulation.

Our response: We appreciate the reviewer’s comment. We have added Bode plots of the hydrogel and the PBS solution as supplementary Fig 12 (c). In supplementary Fig. 12, we chose the frequency value of 100 kHz for the calculation of the conductivity of the hydrogel and PBS because the impedance value measured at frequency ranges higher than the characteristic frequency (that is, $1/2\pi RC$) gives the correct resistance value of the bulk solution, in this case, the hydrogel and PBS. According to the Bode plots of the hydrogel and PBS solution, the characteristic frequency was estimated to be ~ 1 kHz for the hydrogel and ~ 2 kHz for the PBS. Therefore, we calculated the conductivity using the impedance value of the hydrogel measured at 100 kHz which is much higher than 2 kHz^{62, 63}.

We also agree with the reviewer’s comment that we should compare the impedance of the hydrogel and PBS at lower frequency. As shown in supplementary Fig. 12 (c), at 1 kHz, the impedance of PBS and hydrogel were $19.238 \pm 2.787 \Omega$ and $24.354 \pm 0.193 \Omega$, respectively, which showed rather similar values than those in the high frequency (100 kHz).

Regarding the usage of ‘conductive’ word for the hydrogel, we intended to emphasize that hydrogel still allowed for neural signal transmission due to its ion transport properties although it showed 55% of the conductivity level in PBS. It is true that the introduction of hydrogel increases the impedance of the neural electrode as shown in Fig. 2j. Therefore, in this study, we addressed this issue by utilizing the Au-blackened electrodes. Figure 2j compared electrochemical impedance spectra graph of various neural interface electrodes. Although the electrochemical impedance of the hydrogel-coated neural electrode was higher than that of the neural electrode without hydrogel coating due to the lower conductivity of the hydrogel than PBS solution, it was still six times lower than the impedance of the bare neural electrode without Au black. As a result, the electrical performance of the entire neural electrode was overall improved. However, we agree with the reviewer that the usage of ‘conductive’ word can mislead the reader. Therefore, we removed the ‘conductive’ word in Fig. 1c, Supplementary Fig. 12 and in the manuscript.

Our modification to the manuscript: We have added the additional graph in supplementary Fig. 12 (c).

Supplementary Fig. 12

Supplementary Fig. 12 | Nyquist plots of the PBS (a) and hydrogel (b) (n=3). The range of frequency range was from 1 Hz to 100 kHz and inset graph shows same Nyquist plot data in the high-frequency range (from a few kHz to 100kHz). (c) Bode plots of the PBS and hydrogel.

Page 8, line 292-297 “The measured conductivity of the hydrogel was 0.010188 ± 0.000212 S/cm, still showing 55% of the conductivity level in PBS (0.018478 ± 0.00019 S/cm). In terms of the impedance, the impedance of PBS and hydrogel at 1 kHz were $19.238 \pm 2.787 \Omega$ and $24.354 \pm 0.193 \Omega$, respectively, which showed rather similar values than those in the high frequency range. (Supplementary Fig. 12).”

Added references

62. Park TH, Park S, Yu S, Park S, Lee J, Kim S, *et al.* Highly Sensitive On-Skin Temperature Sensors Based on Biocompatible Hydrogels with Thermoresponsive Transparency and Resistivity. *Advanced Healthcare Materials* 2021, **10**(14): 2100469.
63. Kim S, Park S, Choi J, Hwang W, Kim S, Choi I-S, *et al.* An epifluidic electronic patch with spiking sweat clearance for event-driven perspiration monitoring. *Nature Communications* 2022, **13**(1): 6705.

Minor comment #9: In Supplementary Fig. 9, the authors showed that sensitivity to broad pressure could be improved by introducing a cuboid structure. Did the authors compare this cuboid structure with other structures with different geometries or arrangements such as pyramidal arrays?

Our response: We thank the reviewer for this careful comment. Through the newly added theoretical section (Supplementary Note. 1 and Supplementary Fig. 10) on the cuboid structure effect, we have demonstrated that the structure height is the main factor. However, as you suggested, we conducted a pyramid-shaped structure with the same height as the cuboid structure to explore other possibilities. As a result, as expected, there seemed to be a minor effect, but it did not bring about significant changes.

Our modification to the manuscript:

Supplementary Fig. 10 | A schematic image illustrating the theoretical operating mechanism of the cuboid structure. It displays images of the sensor's deformation with broadly applied

pressure when a film-type sensor with a cuboid structure is embedded in soft material.

Supplementary Note 1

The theory of the cuboid structure effect.

Considering the tactile sensor embedded in soft skin tissue or CFAS, it is possible to simplify this state as shown in Supplementary Fig 10. E_{rigid} and E_{soft} are Young's modulus of the cuboid structure and the surrounding soft material, respectively. In our research, E_{rigid} is 52.19 MPa as PUA, while E_{soft} can be considered to be around 10 to 1,000 kPa for materials like skin, CFAS, or other soft biological tissues. When pressure with a flat and wide object is applied by the displacement of δ , the degree of compression varies depending on the presence of the cuboid structure. Here, we considered that the rigid cuboid structure has not compressed as Young's modulus of the cuboid structure is about 1,000 times higher than the CFAS in our experiment. Therefore, the compressed strain of the surroundings can be expressed as follows when we consider the sensor's thickness, t , to be negligible:

$$\varepsilon_{h=0} = \frac{\delta}{H_0} \quad (1)$$

$$\varepsilon_{h \neq 0} = \frac{\delta}{H_0 - h} \quad (2)$$

H_0 is the thickness of the surroundings before pressure was applied, and h is the cuboid structure height. If we compare the two compressive strains, $\varepsilon_{h \neq 0}$ is always higher than $\varepsilon_{h=0}$. It means surroundings where cuboid structure exists, get compressed more by higher concentrated stress as the surroundings are the same materials. In addition, this phenomenon becomes more pronounced as h increases due to the two comparison equations.

Supplementary Fig. 18 | The effects based on different structural forms. **a**, Tactile sensors with cuboid and pyramid structures. **b**, Resistance change graph embedded in CFAS.

Page 9, line 362-366 “Additionally, structures other than the cuboid were investigated, as shown in Supplementary Fig 18. As theorized earlier, since the height of the structure is the most critical factor for the bending effect, there was not a significant difference in the sensor's performance between two structures which has the same height.”

Minor comment #10: In Line 281, the difference between an Au-blackened Au electrode and Au electrode should be explained, along with a background introduction of why an Au-blackened Au electrode has better electrical performance.

Our response: We appreciate the reviewer’s comment. In this study, electrodeposition of the Au-black onto the Au electrode formed an Au nanogranular surface, leading to an increase in the electrode electrochemical surface area (ESA) (figure 2h)⁴⁰. The benefits of the high ESA are a reduction in impedance and an increase in charge storage capacity (CSC) indicating high charge injection capacity for effective stimulation⁴¹.

To validate this, we measured the CSC of the Au-blackened Au electrode and bare Au electrode. The CSC of the Au-blackened Au electrode was five times higher than that of the bare Au electrodes, indicating that the electrodeposition of black Au increased the ESA and improved the electrical performance of the electrode (Supplementary Fig. 14). We have added more background introduction of the effect of the Au-blackened Au electrode in the manuscript as follows.

Our modification to the manuscript:

Page 7, line 298-301 “The electrodeposition of the Au-black onto the Au electrode formed an Au nanogranular surface, leading to an increase in the electrochemical surface area (ESA) of the electrode⁴⁰ (figure 2h). The increased ESA can reduce the impedance and increase the charge storage capacity (CSC)⁴¹.”

Added references

40. Cogan SF. Neural Stimulation and Recording Electrodes. *Annual Review of Biomedical Engineering* 2008, **10**(1): 275-309.
41. Harris AR, Newbold C, Carter P, Cowan R, Wallace GG. Using Chronopotentiometry to Better Characterize the Charge Injection Mechanisms of Platinum Electrodes Used in Bionic Devices. *Frontiers in Neuroscience* 2019, **13**.

Minor comment #11: In Line 428, the figure number is missing in the parenthesis.

Our response: We thank the reviewer for this careful review. We have added the missing figure number in Line 455.

Our modification to the manuscript:

We have modified the following sentences in the main text.

Page 11, line 450-452 “Collagen layer formations, macrophage expression rate, and degr

ee of apoptosis as a result of *in vivo* implantation were confirmed through histology analysis and IF staining (Fig. 4e-h).”

Reviewer #2

General Comment #1: The paper “Bionic Artificial Skin with a Fully Implantable Wireless Tactile Sensory System for Wound Healing and Restoring Skin Tactile Function” demonstrates a novel approach to replace permanently damaged tactile sensory and accelerate regeneration of severely damaged skin. The complete device in this research with good biocompatibility and effective nerve stimulation properties is impressive, so this review recommends to publish the article in the journal, with some minor revisions below.

Our response: We thank the reviewer for this positive comment, and the recommendation to publish in *Nature Communications*. We made our revision with pleasure based on the reviewer’s opinion, and all the details of our modifications are indicated in our responses.

Comment #1: There is no fully completed device figure with CFAS in the main figures. It would be better if you replace Fig. 2a with WTSA of Supplementary Fig. 4.

Our response: We thank the reviewer for this vulnerable comment. As reviewer recommended, we have replaced the Fig. 2a with the completed integrated device figure with CFAS.

Our modification to the manuscript:

Fig. 2a

Comment #2: There is no mention of whether the experiments in Fig. 2d,e are the result of the tactile sensor being inside the CFAS. If they are, it would be better to indicate the ‘CFAS’ in the text or in the figure.

Our response: We appreciate the reviewer’s feedback. Unfortunately, Figures 2d, e represents the experiment simply showing the role of the cuboid structure by embedding them in skin modulus like soft PDMS. However, the results performed inside the CFAS are presented in Figure 3f. In order to reduce any potential confusion, we have added the following sentence to the manuscript.

Our modification to the manuscript:

Page 7, line 254-257 “Figure 2d,e present the tactile sensor characteristics, embedded in a skin-like soft PDMS, depending on the presence of the cuboid structure under a localized and broadly applied pressure up to 100 kPa, which corresponds well to the FEM simulation results.”

Comment #3: I think the correlation between the dimension and effect of the cuboid structure is lacking. Is there no correlation between length or width?

Our response: Thank you for pointing out this issue. We believe that the height of the cuboid structure is the key factor for the bending effect. To provide an explanation, we have included a theoretical section in Supplementary Note. 1 and Supplementary Fig. 10 for clarification.

Our modification to the manuscript:

Page 7, line 265-268 “Supplementary Fig. 10 and Supplementary Note. 1 theoretically demonstrate this effect. As shown, in the presence of the cuboid structure, there is more compression along with stress concentration in the surrounding areas, with the height of the cuboid structure being a key factor.”

Supplementary Fig. 10 | A schematic image illustrating the theoretical operating mechanism of the cuboid structure. It displays images of the sensor's deformation with broadly applied pressure when a film-type sensor with a cuboid structure is embedded in soft material.

Supplementary Note 1

The theory of the cuboid structure effect.

Considering the tactile sensor embedded in soft skin tissue or CFAS, it is possible to simplify this state as shown in Supplementary Fig. 10. E_{rigid} and E_{soft} are Young's modulus of the cuboid structure and the surrounding soft material, respectively. In our research, E_{rigid} is 52.19 MPa as PUA, while E_{soft} can be considered to be around 10 to 1,000 kPa for materials like skin, CFAS, or other soft biological tissues. When pressure with a flat and wide object is applied by the displacement of δ , the degree of compression varies depending on the presence of the cuboid structure. Here, we considered that the rigid cuboid structure has not compressed as Young's modulus of the cuboid structure is about 1,000 times higher than the CFAS in our experiment. Therefore, the compressed strain of the surroundings can be expressed as follows when we consider the sensor's thickness, t , to be negligible:

$$\varepsilon_{h=0} = \frac{\delta}{H_0} \quad (1)$$

$$\varepsilon_{h \neq 0} = \frac{\delta}{H_0 - h} \quad (2)$$

H_0 is the thickness of the surroundings before pressure was applied, and h is the cuboid structure height. If we compare the two compressive strains, $\varepsilon_{h \neq 0}$ is always higher than $\varepsilon_{h=0}$. It means surroundings where cuboid structure exists, get compressed more by higher concentrated stress as the surroundings are the same materials. In addition, this phenomenon becomes more pronounced as h increases due to the two comparison equations.

Comment #4: In the fig. 2d,e, the results seem to be explained well, but the explanation of the principle seems to be lacking. If you can add an image of the principle, it will be helpful to understand.

Our response: We appreciate the reviewer’s comments. We hope the response for this comment would have been replaced by the points we have provided in the reviewer’s third comment (**Comment #3**), as they adequately address the matter.

Comment #5: Please add a scale bar to Supplementary Fig. 9b.

Our response: As you recommended, we have added the previously omitted longitudinal strain scale bar.

Our modification to the manuscript:

Supplementary Fig. 9 | a, Schematic illustration providing an overview of the experimental setup for the tactile sensor property test. w signifies the width of the rod, while h represents the height of the cuboid structure. **b**, FEM longitudinal strain results under different conditions. It compares sensors without and with a cuboid structure, denoted as $h = 0 \mu\text{m}$ (black boxes) and

$h = 500 \mu\text{m}$ (blue boxes), respectively. The sensors were pressed by a rod with different diameters, namely $w = 2 \text{ mm}$ and $w = 20 \text{ mm}$. In the absence of a cuboid structure, it can be confirmed that even when pressure is clearly applied over a wide area, such as $w = 20 \text{ mm}$, no strain is formed on the sensor

Comment #6: Regarding the experiment in Supp. Fig. 9b, the $h=500 \text{ w}=2$ case should have more force concentrated at the tip than the $h=0 \text{ w}=2$ case, which would fit your logic, but isn't the data the opposite?

Our response: We thank the reviewer for this careful comment. As the reviewer pointed out, there were inaccuracies in my writing, so we have made the following revisions to the manuscript.

Our modification to the manuscript:

Page 7, line 250-254 “Therefore, the tactile sensor with the cuboid structure (Supplementary Fig. 9; Blue box) effectively bends and shows almost identical stress distribution under localized and broadly applied pressure while tactile sensor without the cuboid structure (Supplementary Fig. 9; Black box) only bends when localized pressure is applied and merely bends under broad-area pressure.”

Reviewer #3

General Comment #1: The manuscript presents a sophisticated wireless-powered tactile sensory system embedded in artificial skin (WTSA). This innovation aims to restore tactile function and accelerate wound healing in severe skin damage caused by burns, accidents, or other traumas. The tactile sensing involves perceiving external pressures and translating them into frequency-based signals. Subsequently, these signals trigger nerve stimulation, effectively emulating tactile sensations and inducing leg movements. The wound healing effect is designed by incorporating collagen and fibrin-based artificial skin (CFAS). The entire system is implantable with flexible-PCB-based embedded system capable of wireless power transfer. The concept of a fully implantable wireless sensory system appears to be novel. Integrating distinct elements to reestablish tactile sensation and expedite wound healing may offer a unique avenue to address skin damage.

Our response: We thank the reviewer for summing up our manuscript and giving positive comments.

General Comment #2: It is less clear though what the key fundamental innovation here is beyond the sophisticated system integration, which certainly is a solid achievement. In addition, this reviewer also has the following concerns regarding the significance of this work and its detailed research approach.

Our response: For severe skin damage caused by burns, accidents, or other traumas which permanently dysfunction the tactile sensing, not only tactile sensory replacement but also wound regeneration should be considered. However, previous studies only focus on either restoring the tactile sensory or promoting wound regeneration, lacking a comprehensive solution for severe skin damage with impaired tactile function.

Our single substrate integrated implantable WTSA device represents a new paradigm for replacing the tactile sensory function and simultaneously regenerating full-thickness skin to replace a severely damaged skin area. By integrating the collagen and fibrin-based artificial skin (CFAS) which has similar property of extracellular matrix (ECM), the WTSA accelerates layer-by-layer skin regeneration by providing microenvironments that mimic the biological and mechanical properties of the native skin.

The WTSA mimics the tactile sensing mechanism by converting the intensity of applied pressure measured through tactile sensor, embedded in artificial skin to frequency-based sawtooth pulse signal, which is capable of stimulating the sciatic nerve with conductive gelatin coated nerve interface, minimizing the foreign body reaction enabling stable long-term electrical stimulation. Therefore, our WTSA represents a fundamental innovation by accelerating the layer-by-layer skin regeneration and replacing the permanently damaged tactile sensory simultaneously while previous studies only focus on either skin regeneration or replacing the tactile function. Our sophisticated integrated WTSA is crucially important since it represents the potential to be applied to biomimicking artificial sensory systems for various incurable nerve damage applications.

Comment #1: To add into the significance of the work, the authors are recommended to discuss how many patients are affected by severe skin damage, which also leads to permanent tactile sensory impairment.

Our response: We thank the reviewer for this careful comment. We have discussed how many patients are affected by the severe skin damage as reviewer recommended. In the United States alone, over 400,000 patients are suffering from burn related injuries, and severe skin damage occurs can be caused by diverse factors including accidents, traumas, and other unfortunate incidents. Severe skin damage commonly results in damage to mechanoreceptors and their associated nerves, consequently leading to widespread instances of tactile loss and affliction for individuals across the worldwide. For better readership, we added the following sentences in the manuscript.

Our modification to the manuscript:

We have added the following sentences in the main text.

Page 3, line 101-104, “In the United States alone, over 400,000 patients are suffering from burn related injuries annually and more patients are considered to suffer from severe skin damage, highlighting the widespread nature of this issue and its impact on tactile loss globally.”

Added references

1. Evans CS, Hart K, Self WH, Nikpay S, Thompson CM, Ward MJ. Burn related injuries: a nationwide analysis of adult inter-facility transfers over a six-year period in the United States. *BMC Emergency Medicine* 2022, **22**(1): 147.

Comment #2: The tactile sensing has just one sensor, with no spatial resolution. This is quite different from normal skin, where there is a high density of mechanoreceptors. It is hard to imagine how the proposed WTSA can scale up to many sensors.

Our response: We thank the reviewer for this careful comment. As mentioned by the reviewer, the skin is composed of mechanoreceptors with high density which has high spatial resolution. Regarding the proposed WTSA in this study, the crack-based tactile sensor has the potential to be created as an array, thus manifesting as a multi-channel tactile sensor. Furthermore, the components of the WPPFM including the ring oscillators, rectifiers and related passive elements, responsible for modulating tactile signals and wireless power supply, can be produced through an integrated circuit (IC) chip, facilitating the incorporation of a substantial quantity of ring oscillators within a restricted size. Therefore, by integrating the multi-channel crack-based tactile sensor and manufactured IC chip, our proposed WTSA can be scaled up to have high spatial resolution.

Our modification to the manuscript: None

Comment #3: Neural stimulation is typically charge-balanced, while this paper uses a single

phase, which can lead to safety concerns.

Our response: We thank the reviewer for this careful comment. In the case of nerve stimulation using conventional external electrical stimulation equipment, charge-balanced neural stimulation is possible. However, in this study external electrical stimulation equipment usage was impossible because WTSA is wireless powered implantable device. We designed the circuit using ring oscillator to modulate the resistance based tactile signal into frequency based sawtooth signal for effective signal modulation and neural stimulation^{7, 9, 10}. Our signal modulation circuit can precisely modulate the different intensity of applied pressure by generating frequency based sawtooth signal. If additional circuit which converts single phase output into charge-balanced signal is added, the size of the device can be bulky which is not appropriate for implantable device. The additional circuit may cause the performance degradation of signal modulation and distortion of the output signal due to the unintended noise occurrence. In addition, some of the recently reported studies uses single phase signal for neural stimulation^{7,8,9}. However, we agree with the reviewer's comment that the charge-balanced neural stimulation can minimize the safety concerns and we are planning to design the circuit with change-balanced output signal for the future work.

Our modification to the manuscript: To clarify this point, we have added the following references in the main text.

Page 5, line 202-206 "Then WPPFM convert the tactile signal in form of resistance to voltage difference via voltage divider, and ring oscillator modulate the voltage difference into sawtooth pulse signal with different frequencies followed by sciatic nerve stimulation through hydrogel coated neural interface electrodes^{7, 9, 10}."

Added references

7. Kim Y, Chortos A, Xu W, Liu Y, Oh JY, Son D, *et al.* A bioinspired flexible organic artificial afferent nerve. *Science* 2018, **360**(6392): 998-1003.
9. Lee Y, Liu Y, Seo D-G, Oh JY, Kim Y, Li J, *et al.* A low-power stretchable neuromorphic nerve with proprioceptive feedback. *Nature Biomedical Engineering* 2023, **7**(4): 511-519.
10. Wang W, Jiang Y, Zhong D, Zhang Z, Choudhury S, Lai J-C, *et al.* Neuromorphic sensorimotor loop embodied by monolithically integrated, low-voltage, soft e-skin. *Science* 2023, **380**(6646): 735-742.

Comment #4: The nerve stimulation is applied to the efferent nerve that drives the leg movement, but the tactile sensing is done through the afferent nerve. It is unclear how the demonstration done in this work can be effective for the recovery of tactile sensing.

Our response: We thank the reviewer for this valuable comment. As the reviewer commented, efferent nerves are associated with leg movement, and afferent nerves are associated with tactile sensing. In this work, permanently impaired tactile sensory resulting from the severe skin damage, which often damages the mechanoreceptors and associated afferent nerves, can be replaced by our WTSA. The crack-based tactile sensor of the WTSA sense the externally

applied pressure and the sciatic nerve is stimulated with modulated tactile signal in form of frequency based sawtooth signal. Our demonstration validates the effective nerve stimulation achieved through the modulation of tactile signals in response to the different intensity of applied pressure in the aspect of neural interfacing.

Our modification to the manuscript: None

Comment #5: The wound healing process doesn't seem to be significantly affected by the CFAS. The claim of accelerating wound healing may be premature.

Our response: As the reviewer said, the expression accelerating wound healing seems premature. So this expression was changed to promote wound healing when explaining the healing effects of CFAS in manuscript. In addition, although CFAS may not significantly affect wound regeneration compared to defect, CFAS was used because it alleviated the inhibition of wound healing when the tactile sensor was used alone and helped wound regeneration. Also, there was an omitted part in the graph (Fig. 3e) of the previous manuscript, so it was revised in the manuscript.

Our modification to the manuscript: To clarify this point we have modified the manuscript by changing the word 'accelerate' to 'promote' and whether there was any exaggerated expression, it was thoroughly reviewed and corrected.

REVIEWERS' COMMENTS

Reviewer #1 (Remarks to the Author):

The authors have substantially improved their manuscript and provided comprehensive explanations for all the questions in the first round of review. This manuscript is now ready to be published in its current form.

Reviewer #2 (Remarks to the Author):

The authors successfully addressed all the issues the reviewer asked, so this reviewer recommend Nature Communications to publish this article.

Reviewer #3 (Remarks to the Author):

The authors have effectively addressed most of this reviewer's previous comments. For wound healing in particular, although on Day 21 there is no significant difference between the e-skin and controls, this reviewer agrees that the difference is prominent in the early days (e.g., Day 3), which is important, especially considering the patient experience. This result is indeed a great achievement.

The authors are still suggested to add a short discussion in their manuscript to comment on the scalability of their tactile sensing. However, there is no need for another review from this reviewer.

Reviewer #1

General Comment #1: The authors have substantially improved their manuscript and provided comprehensive explanations for all the questions in the first round of review. This manuscript is now ready to be published in its current form.

Our response: We thank the reviewer for this positive comment, and the recommendation to publish in *Nature Communications*.

Reviewer #2

General Comment #1: The authors successfully addressed all the issues the reviewer asked, so this reviewer recommend Nature Communications to publish this article.

Our response: We thank the reviewer for this positive comment, and the recommendation to publish in *Nature Communications*.

Reviewer #3

General comment #1: The authors have effectively addressed most of this reviewer's previous comments. For wound healing in particular, although on Day 21 there is no significant difference between the e-skin and controls, this reviewer agrees that the difference is prominent in the early days (e.g., Day 3), which is important, especially considering the patient experience. This result is indeed a great achievement.

Our response: We thank the reviewer for this positive comment.

Comment #1: The authors are still suggested to add a short discussion in their manuscript to comment on the scalability of their tactile sensing. However, there is no need for another review from this reviewer.

Our response: We thank the reviewer for this comment and we added the short discussion about scalability of tactile sensing for better readership.

Our modification to the manuscript: We added the discussion in Discussion section of the revised manuscript.

Page 13, line 527-529 “Also, by developing such a device into a multi-channel tactile sensing and converting units, it can provide fundamental treatment for severe skin damage over a large area.”